# Integrated relationship of nasopharyngeal airway host response and microbiome associates with bronchiolitis severity

Michimasa Fujiogi [1] ✉, Yoshihiko Raita [1], Marcos Pérez-Losada[2,3], Robert J. Freishtat[4,5,6], Juan C. Celedón [7], Jonathan M. Mansbach[8], Pedro A. Piedra[9], Zhaozhong Zhu [1], Carlos A. Camargo Jr. [1] & Kohei Hasegawa [1]

Bronchiolitis is a leading cause of infant hospitalizations but its immuno-pathology remains poorly understood. Here we present data from 244 infants hospitalized with bronchiolitis in a multicenter prospective study, assessing the host response (transcriptome), microbial composition, and microbial function (metatranscriptome) in the nasopharyngeal airway, and associate them with disease severity. We investigate individual associations with disease severity identify host response, microbial taxonomical, and microbial functional modules by network analyses. We also determine the integrated relationship of these modules with severity. Several modules are significantly associated with risks of positive pressure ventilation use, including the host-type I interferon, neutrophil/interleukin-1, T cell regulation, microbial-branched-chain amino acid metabolism, and nicotinamide adenine dinucleotide hydrogen modules. Taken together, we show complex interplays between host and microbiome, and their contribution to disease severity.

Bronchiolitis—the most common lower respiratory infection among infants—is an important health problem[1]. While 30–40% of infants develop clinical bronchiolitis, its severity ranges from a minor nuisance to fatal infection[2,3]. Bronchiolitis is also the leading cause of hospitalization in US infants, accounting for ~110,000 hospitalizations annually[4]. Approximately 5% of these infants undergo mechanical ventilation[4]. However, traditional risk factors (e.g., prematurity) do not sufficiently explain the differences in disease severity[3] and its pathobiology remains to be elucidated. Our limited understanding of the disease mechanisms has hindered efforts to develop targeted treatment strategies in this large patient population.

Emerging evidence has pointed out the pathobiological role of respiratory viral pathogens, host response, and microbiome in infant bronchiolitis[3]. Studies have reported *individual* associations of upper airway[5,6] and circulating[7–10] transcriptome, microRNA[11], cytokine[12–16], proteome[10], metabolome[17–20], and microbiota[7,17,21–27] profiles with bronchiolitis severity. However, these findings using *single*-element data were unable to uncover the *integrated* contribution of host response and microbiome to the pathobiology of bronchiolitis.

[1]Department of Emergency Medicine, Massachusetts General Hospital, Harvard Medical School, Boston, MA, USA. [2]Computational Biology Institute, Department of Biostatistics and Bioinformatics, The George Washington University, Washington, DC, USA. [3]CIBIO-InBIO, Centro de Investigação em Biodiversidade e Recursos Genéticos, Universidade do Porto, Campus Agrário de Vairão, Vairão, Portugal. [4]Center for Genetic Medicine Research, Children's National Hospital, Washington, DC, USA. [5]Division of Emergency Medicine, Children's National Hospital, Washington, DC, USA. [6]Department of Pediatrics, George Washington University School of Medicine and Health Sciences, Washington, DC, USA. [7]Division of Pulmonary Medicine, Department of Pediatrics, UPMC Children's Hospital of Pittsburgh, University of Pittsburgh, Pittsburgh, PA, USA. [8]Department of Pediatrics, Boston Children's Hospital, Harvard Medical School, Boston, MA, USA. [9]Departments of Molecular Virology and Microbiology and Pediatrics, Baylor College of Medicine, Houston, TX, USA. ✉e-mail: fujiogi-stm@umin.ac.jp

Despite their clinical and research significance, no study has integrated host response and microbiome (both composition and function) to determine their interrelationship with disease severity in infants with bronchiolitis.

To address this major knowledge gap, we applied integrated-omics and network approaches to dual-transcriptome data—host response (transcriptome), microbiome composition and its function (metatranscriptome) in the nasopharyngeal airway—from a multi-center prospective cohort of infants hospitalized for bronchiolitis (Fig. 1). First, we examined the *individual* relationship of each omics element with disease severity (positive pressure ventilation [PPV] use) and identified the unique host response, microbiome composition and function signatures. Second, we identified distinct networks (modules) in each omics element—9 host response, 7 microbial composition, and

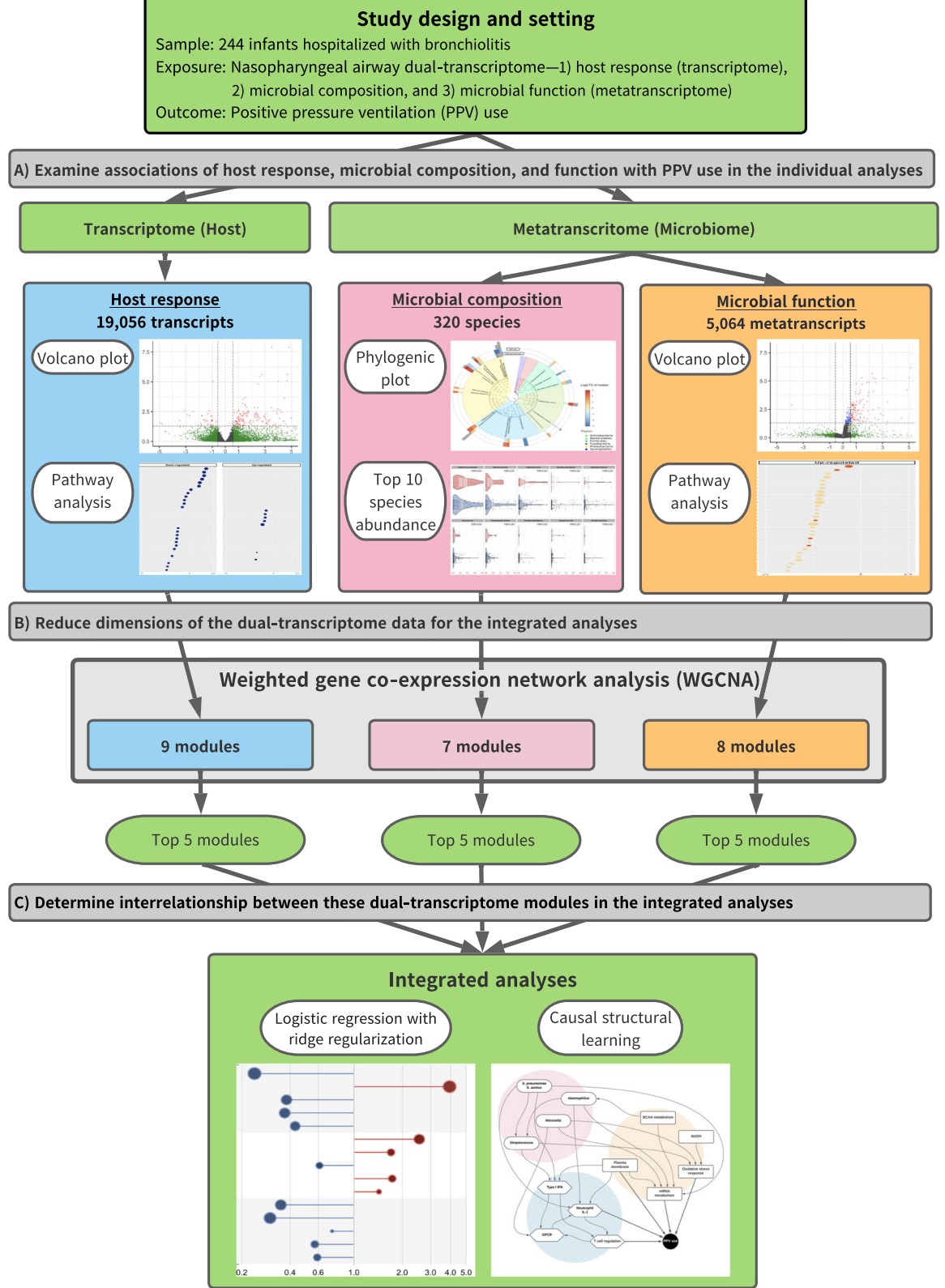

Fig. 1 | **Analytic flow of integrated-omics analysis.** This flowchart presents a brief overview of the main analytical steps in the current study. The steps are shown in order from top to bottom (**A** to **C**). For each of 1) host response (transcriptome), 2) microbial composition (metatranscriptome), and 3) microbial function (meta-transcriptome) data elements, we individually performed the analysis in steps A and B. Then, we subsequently integrated these omics data in step C. **A** We examined the relationship of each omics data element with the risk of PPV use at the individual data level. **B** To reduce the dimensions of the host response, microbial composition, and microbial function data, we performed a weighted gene co-expression network analysis and identified distinct networks (modules). In each omics element, we selected the top five modules with the highest correlation of PPV use and biological significance for the subsequent integrated analyses. **C** Finally, to determine the integrated relationships of these dual-transcriptome modules with the risk of PPV use, we constructed a logistic regression model with ridge regularization. To uncover the causal relationship structure between these dual-transcriptome modules, we also applied a causal structural learning approach. Abbreviations: FDR, false discovery rate; PPV, positive pressure ventilation; WGCNA, weighted gene co-expression network analysis.

8 microbial function modules—that have distinct biological and microbial characteristics. Finally, we examined their *integrated* relationship with the PPV risk and identified that several modules were associated with bronchiolitis severity, including the host-type I interferon (IFN), neutrophil/interleukin (IL)−1, T-cell regulation, *Streptococcus pneumoniae/Staphylococcus aureus*, and microbial-branched-chain amino acid (BCAA) metabolism, and nicotinamide adenine dinucleotide hydrogen (NADH) modules.

## Results

### Baseline characteristics

We analyzed data from a multicenter prospective cohort study of infants hospitalized for bronchiolitis—the 35th Multicenter Airway Research Collaboration (MARC-35) study. This study enrolled 1,016 infants (age < 1 year) with bronchiolitis at 17 sites across 14 US states (Supplementary Table 1) over three bronchiolitis seasons[28]. The current study included 244 infants who were randomly selected for nasopharyngeal airway dual-transcriptome testing (Supplementary Fig. 1). The analytic and non-analytic cohorts did not significantly differ in the baseline characteristics (*P* ≥ 0.05; Supplementary Table 2), except for daycare use and RSV infection. Among the analytic cohort, the median age was 3 (IQR, 2−6) months, 40% were female, and 42% were non-Hispanic white (Table 1). Overall, 91% of study participants had RSV infection, 21% had rhinovirus (RV) infection, and 12% had RSV/RV coinfection. During hospitalizations for bronchiolitis, 7% of participants underwent PPV and 17% received intensive care treatment (defined by PPV use and/or admission to the intensive care unit).

### Individual relationships of nasopharyngeal airway host transcripts, microbial composition, and function with disease severity

Of 19,056 host transcripts detected in the nasopharyngeal airway of infants with bronchiolitis, 197 were significantly associated with the risk of PPV use (Benjamini–Hochberg false discovery rate [FDR] of <0.05 and ≥|1.5|-fold change; Fig. 2A). In the functional pathway analysis of Gene Ontology (GO) biological process, infants with PPV use had 102 differentially enriched pathways (FDR < 0.05)—e.g., down-regulated type I IFN, IFN-γ, virus defense response, and T-cell activation pathways as well as upregulated neutrophil pathways, compared to those without PPV use (Fig. 2B). The differentially enriched pathways in the GO molecular function (e.g., downregulated NADH dehydrogenase pathways) and cellular component (e.g., downregulated major histocompatibility complex [MHC] class II protein complex, and upregulated secretary granule pathways) domains are shown in Supplementary Fig. 2.

A total of 320 microbial species were detected in the nasopharyngeal airway of infants with bronchiolitis. The overall relationship of the 20 most abundant microbial species with the severity outcomes is shown in Fig. 3A. The 20 most abundant species come from 4 major phyla (Actinobacteria, Bacteroidetes, Firmicutes, and Proteobacteria). In the investigation of the 10 most abundant microbial species (which collectively accounted for 93% of the overall composition), all species were significantly associated with the risk of PPV use (all FDR < 0.001; Fig. 3B). For example, a higher abundance of *S. pneumoniae* and a lower abundance of *Moraxella catarrhalis* were significantly associated

## Table 1 | Patient characteristics of 244 infants hospitalized for bronchiolitis

| | Overall (*n* = 244) |
|---|---|
| **Characteristics** | |
| Age (month), median (IQR) | 3.1 (1.7–6.2) |
| Female sex | 98 (40) |
| Race/ethnicity | |
| Non-Hispanic white | 102 (42) |
| Non-Hispanic black | 57 (23) |
| Hispanic | 76 (31) |
| Other or unknown | 9 (4) |
| Maternal smoking during pregnancy | 34 (14) |
| C-section delivery | 84 (34) |
| Prematurity (<37 weeks) | 47 (19) |
| Mostly breastfed for the first 3 months of age | 115 (47) |
| Previous breathing problems before the index hospitalization[a] | |
| 1 episode | 30 (12) |
| ≥2 episodes | 10 (4) |
| History of eczema | 31 (13) |
| Ever attended daycare | 71 (29) |
| Corticosteroid use before the index hospitalization | 18 (7) |
| Lifetime history of systemic antibiotic use | 79 (32) |
| **Clinical presentation** | |
| Body weight at presentation (kg), median (IQR) | 6.1 (4.6–8.0) |
| Respiratory rate at presentation (per minute), median (IQR) | 48 (40–60) |
| Oxygen saturation at presentation | |
| <90% | 18 (7) |
| 90–93% | 29 (12) |
| ≥94% | 190 (78) |
| **Respiratory virus** | |
| Any RSV | 222 (91) |
| Any rhinovirus | 51 (21) |
| RSV/rhinovirus coinfection | 29 (12) |
| Other coinfection pathogens[b] | 47 (19) |
| **Clinical outcomes** | |
| Positive pressure ventilation during hospitalization[c] | 18 (7) |
| Intensive care use during hospitalization[d] | 42 (17) |
| Hospital length of stay (day), median (IQR) | 2 (1–3) |

*IQR* interquartile range, *RSV* respiratory syncytial virus.
Data are *n* (%) of infants unless otherwise indicated. Percentages may not equal 100 because of rounding and missingness.
[a]Defined as an infant having a cough that wakes him or her at night or causes emesis, or when the child has wheezing or shortness of breath without cough.
[b]Adenovirus, bocavirus, *Bordetella pertussis*, enterovirus, human coronavirus NL63, OC43, 229E, or HKU1, human metapneumovirus, influenza A or B virus, *Mycoplasma pneumoniae*, and para-influenza virus 1–3.
[c]Defined as use of invasive and/or non-invasive mechanical ventilation (e.g., continuous positive airway pressure ventilation).
[d]Defined as use of positive pressure ventilation and/or intensive care unit admission.

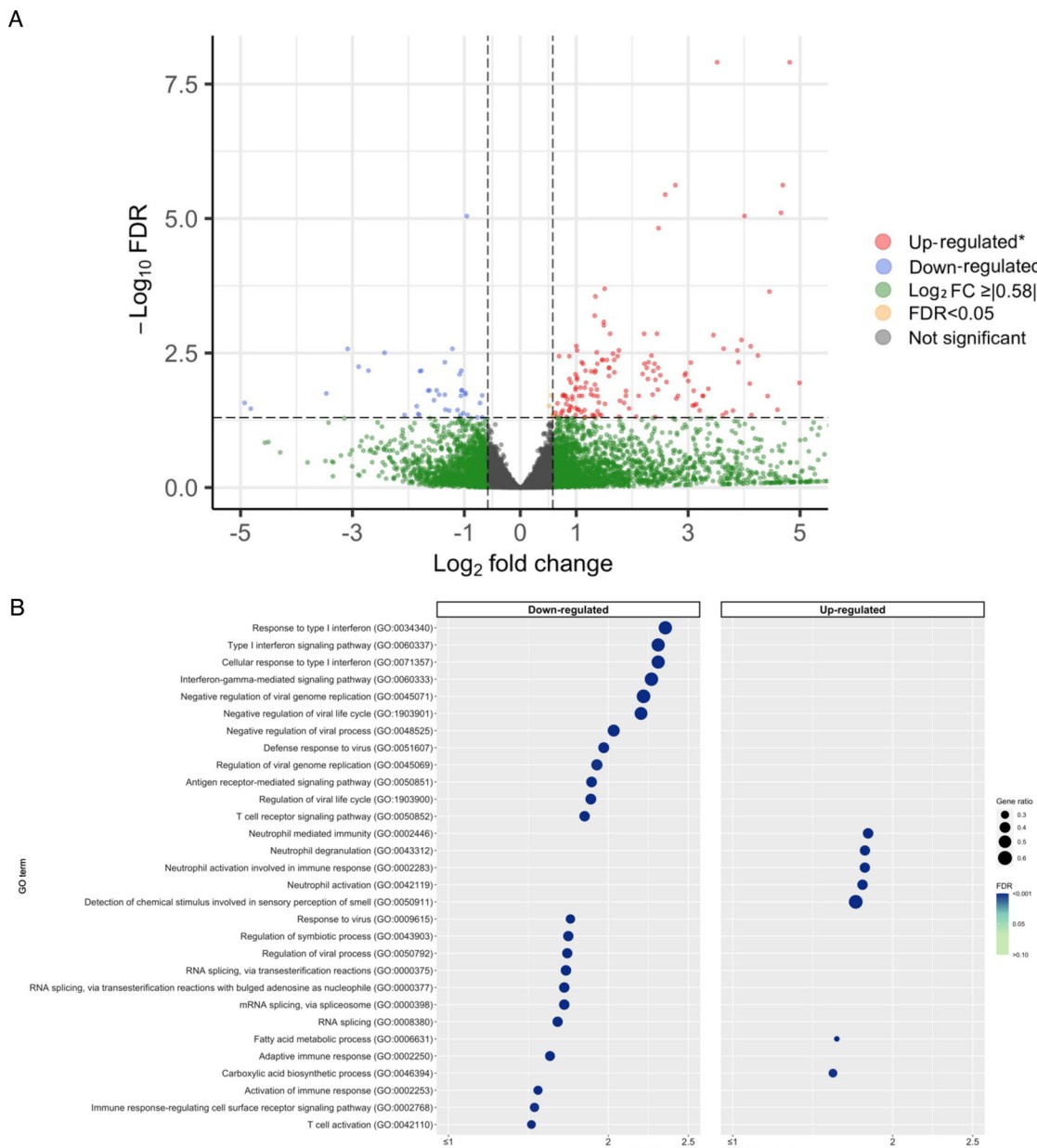

**Fig. 2 | Differential gene expression analysis of host transcriptome data with regard to the use of positive pressure ventilation in infants hospitalized for bronchiolitis. A** Volcano plot of differentially expressed genes (transcriptome). The threshold of log2 fold change is |0.58| (i.e., ≥|1.5|-fold change), and that of FDR is <0.05. There were 197 differentially expressed transcripts that met these criteria. **B** Gene set enrichment analysis (transcriptome). We showed 30 host pathways (GO biological process) with the most significant FDR in the gene set enrichment analysis (GSEA) with downregulated pathways on the left side and upregulated pathways on the right side. We also showed the absolute normalized enrichment score, FDR, and the gene ratio for the corresponding pathways. Abbreviations: FDR false discovery rate, GO gene ontology, GSEA gene set enrichment analysis, PPV positive pressure ventilation.

with the PPV risk. Additionally, a total of 340 fungal species were detected. Of 10 most abundant species, 9 species were significantly associated with the PPV risk (FDR < 0.001; Supplementary Fig. 3). For example, a higher abundance of *Malassezia restricta* was significantly associated with a higher PPV risk (FDR < 0.001).

Of 5064 microbial transcripts detected in the nasopharyngeal airway of infants with bronchiolitis, 129 were significantly associated with the risk of PPV use (FDR < 0.05 and ≥|1.5|-fold change; Fig. 4A). In the functional pathway analysis of GO biological process, infants with PPV use had 5 differentially enriched pathways (FDR < 0.05)—e.g., upregulated lipid metabolism and oxidant detoxification pathways (Fig. 4B). The differentially enriched pathways in the GO molecular function (e.g., upregulated NADH oxidoreductase and antioxidant pathways) and cellular component (e.g., upregulated NADH

dehydrogenase complex pathway) domains are shown in Supplementary Fig. 4.

## Identification of dual-transcriptome modules with distinct biological function

By using differentially enriched host transcripts, microbial species, and microbial function data, the network analysis (weighted gene co-expression network analysis [WGCNA][29]) identified 9 distinct host response (e.g., T-cell regulation, neutrophil/IL-1, type I IFN modules), 7 distinct microbial composition (e.g., *S. pneumoniae/S. aureus* module), and 8 microbial function (e.g., BCAA metabolism, oxidative stress response, NADH modules) modules (Supplementary Tables 3–5). Each of the identified modules was characterized by distinct host biological pathways (Supplementary Table 3), microbial

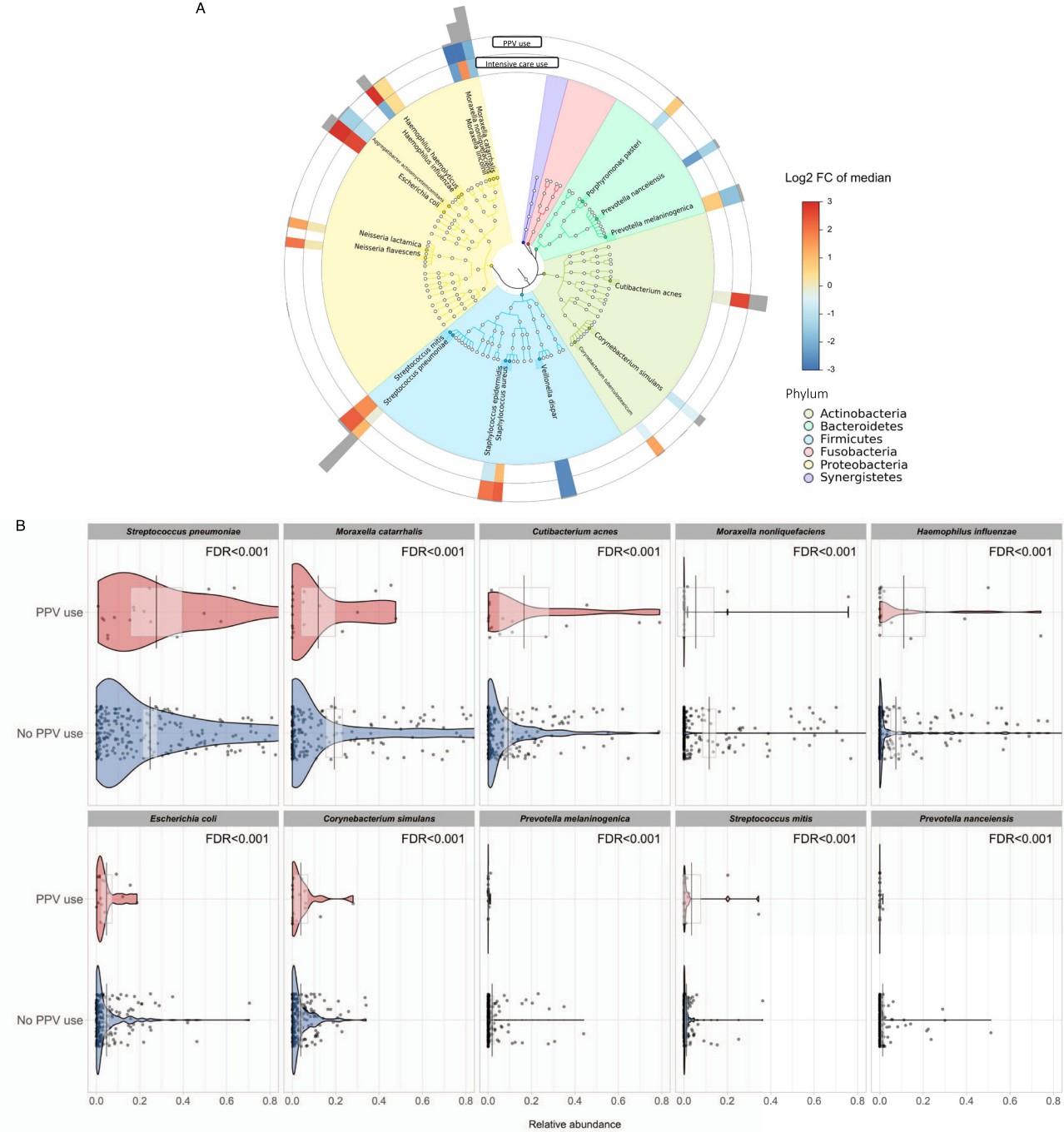

**Fig. 3 | Relationship of abundant microbial species with the risk of higher severity in infants hospitalized for bronchiolitis. A** Phylogenetic plot of top 20 most abundant microbial species in the nasopharyngeal airway of infants hospitalized for bronchiolitis. The colors in the inner circle annotate the six major phyla. The colors in the two internal rings represent the magnitude of the association between the relative abundance of each species and higher severity (PPV use and intensive care use) outcomes. Greyscale bars on the outside of the circular graph are proportional to the microbial species' mean relative abundance. **B** The pirate plots show the comparison of the distribution of ten most abundant species in the nasopharyngeal microbiome in infants hospitalized for bronchiolitis, according to the PPV use. Each point represents each infant. The gray bar and rectangle represent the mean and 95% confidence interval. In the violin plots, the width represents the probability that infants take on a specific relative abundance. The between-group differences in the abundance were tested by fitting Poisson regression models. *n* = 244 biologically independent samples. Abbreviations: FC fold change, FDR false discovery rate, PPV positive pressure ventilation.

species (Supplementary Table 4), and microbial biological pathways (Supplementary Table 5).

### Integrated relationships of nasopharyngeal airway dual-transcriptome modules with disease severity

The integrated analyses used the top five modules with the highest correlation with PPV use and biological significance from each omics

element (Supplementary Tables 3–5). The eigenvalues (the first principal component) of all host response modules, *S. pneumoniae/S. aureus* module, and all microbial function modules were significantly associated with the risk of PPV use (FDR < 0.05; Fig. 5A). Likewise, in the ridge regression analysis adjusting for potential confounders (age, sex, and respiratory virus), the results were consistent (Fig. 5B). For example, the host-T-cell regulation (adjusted odds ratio [adjOR] 0.24;

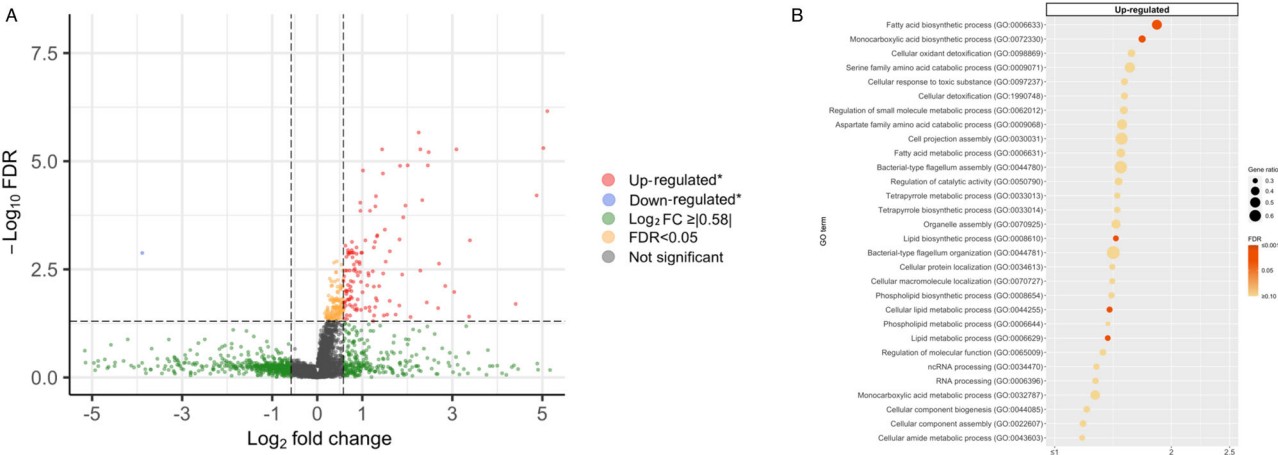

**Fig. 4 | Differential gene expression analysis of microbial function data with regard to the use of positive pressure ventilation in infants hospitalized for bronchiolitis. A** Volcano plot of differentially expressed microbial transcripts (metatranscriptome). The threshold of log2 fold change is |0.58| (i.e., ≥|1.5|-fold change), and that of FDR is <0.05. There were 129 differentially expressed microbial transcripts that met these criteria. **B** Gene set enrichment analysis (GSEA) of the metatranscriptome data. We showed 30 microbial functional pathways (GO biological process) with the most significant FDR in the gene set enrichment analysis (GSEA). Downregulated pathways were not detected. We also showed the normalized enrichment score, FDR, and the gene ratio for the corresponding pathways. Abbreviations: FDR false discovery rate; GO gene ontology; GSEA gene set enrichment analysis.

95% confidence interval [CI] 0.11–0.53), neutrophil/IL-1 (adjOR 3.94; 95% CI 1.70–10.1), and type I IFN (adjOR 0.37; 95% CI 0.14–0.75) modules were significantly associated with the risk of PPV use. Additionally, the *S. pneumoniae/S. aureus* (adjOR 2.55; 95% CI 1.18–5.78), microbial-BCAA metabolism (adjOR 0.73; 95% CI 0.05–0.88), oxidative stress response (adjOR 0.57; 95% CI 0.07–0.78), and NADH (adjOR 0.59; 95% CI 0.06–0.80) modules were significantly associated with the risk of PPV use. In the sensitivity analysis, similar results were observed in the integrated associations with the risk of intensive care use (Fig. 5A and Supplementary Fig. 5). Additionally, in the sensitivity analysis adjusting for race/ethnicity (in addition to age, sex, and virus), the results did not materially change (Supplementary Figs. 6 and 7).

A correlation network (Supplementary Fig. 8) suggests a complex relationship between clinical characteristics, airway microbiome, and host immune responses in the nasopharyngeal airway of infants with bronchiolitis. To uncover the underlying causal relationship between these dual-transcriptome modules, causal structure learning was applied (Fig. 5C). The analysis suggested that, for example, the *S. pneumoniae/S. aureus* module has direct effects on both host-type I IFN and neutrophil/IL-1 modules, which have a subsequent effect on the PPV use through the host-T-cell regulation modules. Additionally, the *S. pneumoniae/S. aureus* module also had subsequent effects on PPV use through microbial-mRNA and oxidative stress response modules.

## Discussion

In this multicenter prospective cohort study of infants hospitalized for bronchiolitis, we first *individually* investigated the relationships of dual-transcriptome data—host response (transcriptome), microbial composition, and microbial function (metatranscriptome)—with disease severity. For example, compared to infants without PPV use, those with PPV use had downregulated host-type I IFN, virus defense response, and T-cell activation pathways as well as upregulated neutrophil pathways. We also found that these infants with higher severity had an increased abundance of *S. pneumoniae* and upregulated microbial-NADH oxidoreductase and antioxidant pathways. Second, we performed the network and integrated-omics analysis. This approach not only demonstrated the modules consistent with the individual-level analyses, but also identified biologically important modules (or networks) that contributed to higher severity. For example, the host-type I IFN, neutrophil/IL-1, T-cell regulation, *S. pneumoniae/S. aureus*, microbial-BCAA metabolism, oxidative stress response,

and NADH modules were significantly associated with the risk of PPV use. To the best of our knowledge, this is the first study that has demonstrated interrelations between host response, microbial composition, and its function in the airway, and their integrated contributions to the disease severity in infants with bronchiolitis.

In agreement with the current study, recent bronchiolitis research has suggested pathobiological roles of respiratory viruses, host response, and microbiome by using *single*-element data—e.g., upper airway[5,6] and circulating[7–10] transcriptome data, and microbiome composition data using 16S ribosomal RNA (16S rRNA) gene sequencing[7,17,21–25,27] or quantitative PCR assay[26]. For example, in a single-center study of 55 infants hospitalized for RSV bronchiolitis using nasopharyngeal transcriptome profiling, Thwaites et al. reported that a lower type I IFN expression was associated with higher severity[6]. In another single-center study of 132 infants with RSV infection using 16S rRNA gene sequencing and whole-blood transcriptome profiling, Piters et al. reported that nasopharyngeal *Streptococcus*-dominated microbiota was associated with overexpression of neutrophil signaling and higher severity[7]. Similarly, in our previous analysis of two cohort studies of infants with bronchiolitis using 16S rRNA gene sequencing, we demonstrated that *Streptococcus*-dominated microbiota profile was associated with a higher risk of intensive care use[21]. Furthermore, our previous integrated-omics analysis of infants with RSV bronchiolitis—which focused on the microbiome taxonomy (i.e., not function), transcriptome, metabolome, and asthma outcome—found that the most-severe endotype (e.g., 19% with PPV use) also had a higher abundance of *S. pneumoniae* and unique host response profile (e.g., low type I interferon response). This endotype also had a non-significantly higher risk of asthma by age 5 years[30]. The current study—applying integrated-omics and network analyses to the dual-transcriptome data—corroborates these prior reports and extends them by demonstrating the integrated relationships of host response, microbial composition, and its function with disease severity in infants with bronchiolitis.

The mechanisms underlying the observed interrelationships warrant clarification. In concordance with our data, studies have suggested the role of host immune response—e.g., type I IFN, neutrophil, and regulatory T cells (Treg)—in the bronchiolitis pathobiology. First, research has shown that RSV infection (specifically with its non-structural 1 and 2 proteins) suppresses induction of type I IFN and IFN-inducible genes, thereby inhibiting innate immune response[31] and that

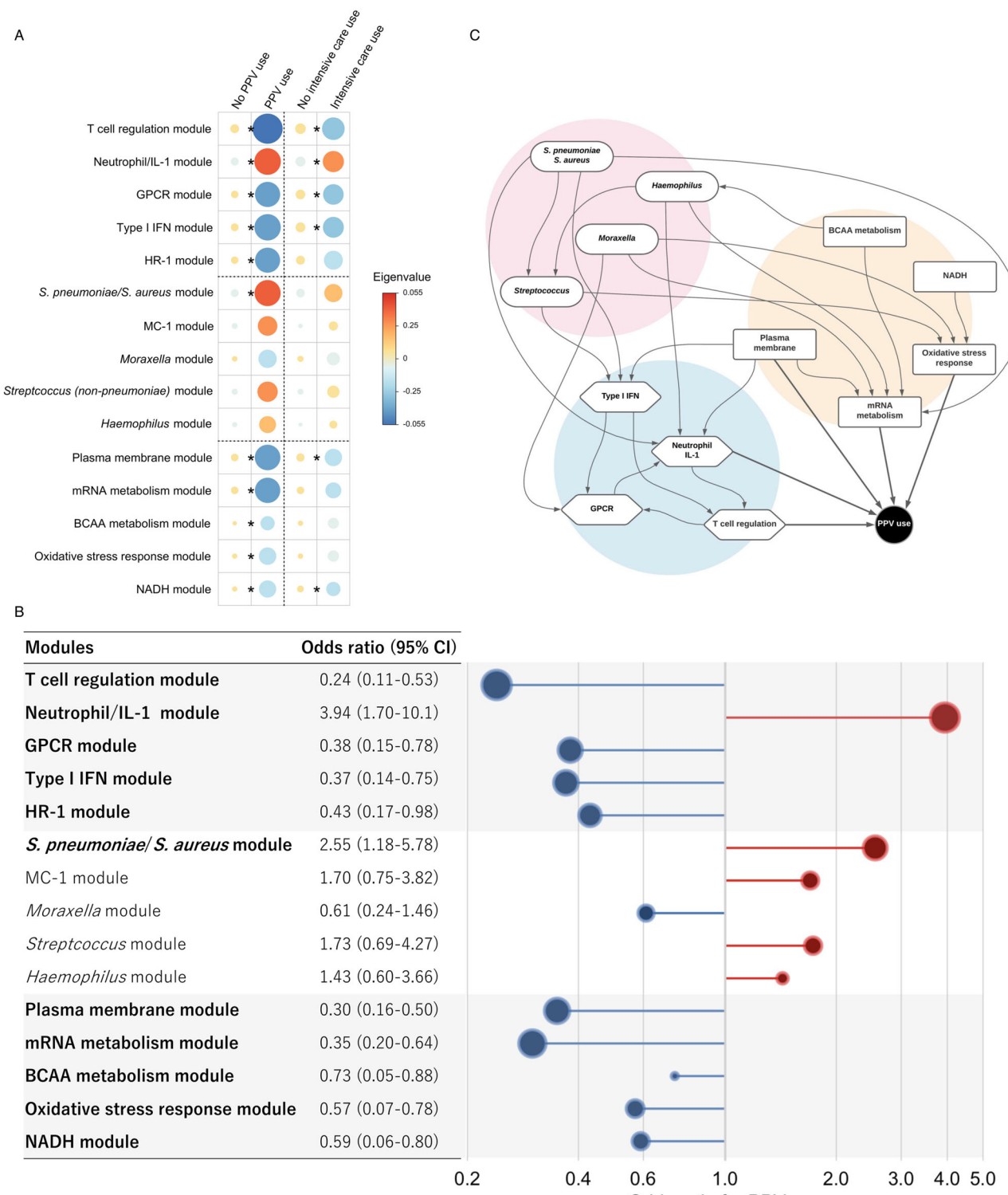

their F protein can also activate IFN-inducible genes with subsequent cell exhaustion of IFNs[32]. Consequently, lower type I IFN level in the airway has been associated with higher disease severity[6,15]. Additionally, a study has also found that type I IFNs are exploited for enhancing immunity against *S. pneumoniae* via regulating innate immune cells[33]. Second, an excessive neutrophil function has been implicated in airway damage and severe bronchiolitis[34]. Neutrophils—the dominant inflammatory cell in the airways of children with bronchiolitis[35–37]—detect virus-associated molecular patterns through their pattern recognition receptors (e.g., toll-like receptors), produce an array of

antimicrobial products (e.g., cathelicidins), and assist the adaptive immune responses[38,39]. Indeed, a previous study has reported an interaction between antimicrobial products and nasopharyngeal airway microbiome composition (e.g., *Streptococcus*-dominance) on the disease severity in infants with bronchiolitis[40]. Third, Tregs have an essential role in ensuring efficient viral clearance by coordinating the recruitment of CD8+ cytotoxic T cells to the airway, controlling innate immune response by neutrophils and NK cells, and limiting an excessive virus-specific T-cell pro-inflammatory response[41]. A previous study revealed that, in infants with severe RSV infection, circulating Tregs

**Fig. 5 | Integrated associations of the dual-transcriptome modules with the use of positive pressure ventilation in infants hospitalized for bronchiolitis.**
**A** Heatmap of the median eigenvalues (the first principal component) for the corresponding modules in each outcome group. The areas of circles and colors represent the median value of the corresponding eigenvalue. The between-group differences tested using two-tailed $t$-test s, accounting for multiple comparisons by applying Benjamini−Hochberg false discovery rate (FDR). Asterisks indicate statistical significance (FDR < 0.05). The exact $P$ values and FDR are the following: In PPV use, T-cell regulation, $P$ value = $7.3 \times 10^{-5}$, FDR = 0.002; Neutrophil/IL-1, $P$ value = $6.7 \times 10^{-3}$, FDR = 0.014; GPCR, $P$ value = $1.1 \times 10^{-2}$, FDR = 0.018; Type I IFN; $P$ value = $7.2 \times 10^{-5}$, FDR = 0.014; HR-1, $P$ value = $2.5 \times 10^{-2}$, FDR = 0.034; *S. pneumonia/S. aureus*, $P$ value = $1.3 \times 10^{-2}$, FDR = 0.020; MC-1, $P$ value = $1.6 \times 10^{-1}$, FDR = 0.197; *Moraxella*, $P$ value = $2.3 \times 10^{-1}$, FDR = 0.244; *Streptococcus*, $P$ value = $2.0 \times 10^{-1}$, FDR = 0.226; *Haemophilus*, $P$ value = $3.8 \times 10^{-1}$, FDR = 0.379; Plasma membrane, $P$ value = $1.1 \times 10^{-17}$, FDR < 0.001; mRNA metabolism, $P$ value = $2.1 \times 10^{-4}$, FDR = 0.001; BCAA metabolism, $P$ value = $6.0 \times 10^{-3}$, FDR = 0.014; Oxidative stress response, $P$ value = $6.4 \times 10^{-5}$, FDR < 0.001; and NADH, $P$ value = $5.5 \times 10^{-5}$, FDR < 0.001. In intensive care use, T-cell regulation, $P$ value = $2.0 \times 10^{-3}$, FDR = 0.030; Neutrophil/IL-1, $P$ value = $6.5 \times 10^{-3}$, FDR = 0.036; GPCR, $P$ value = $1.3 \times 10^{-2}$, FDR = 0.036; Type I IFN, $P$ value = $8.7 \times 10^{-3}$, FDR = 0.036; HR-1, $P$ value = $3.4 \times 10^{-2}$, FDR = 0.064; *S. pneumonia/S. aureus*, $P$ value = $3.4 \times 10^{-2}$, FDR = 0.064; MC-1, $P$ value = $6.2 \times 10^{-1}$, FDR = 0.659; *Moraxella*, $P$ value = $3.4 \times 10^{-1}$, FDR = 0.422; *Streptococcus*, $P$ value = $3.9 \times 10^{-1}$, FDR = 0.448; *Haemophilus*, $P$ value = $7.4 \times 10^{-1}$, FDR = 0.739; Plasma membrane, $P$ value = $1.4 \times 10^{-2}$, FDR = 0.036; mRNA metabolism, $P$ value = $5.0 \times 10^{-2}$, FDR = 0.083; BCAA metabolism, $P$ value = $1.4 \times 10^{-1}$, FDR = 0.211; Oxidative stress response, $P$ value = $2.0 \times 10^{-1}$, FDR = 0.278; and NADH, $P$ value = $1.4 \times 10^{-2}$, FDR = 0.036. **B** Integrated relationship of the dual-transcriptome modules with the risk of PPV use in infants hospitalized for bronchiolitis. The adjusted odds ratio for the outcome was estimated per one unit increased in the eigenvalue (the first principal component) of the corresponding module by fitting a multivariable logistic regression model with ridge regularization. The 95% CIs were estimated by a bootstrap method with 2000 replicates. In the model, we adjusted for age, sex, and respiratory virus. Statistically significant modules are in bold. **C** Causal structural learning is applied to the dual-transcriptomics data. It identifies an underlying causal relationship between these host immune response (blue), microbial species (pink), and microbial function (orange) modules in the niche, and demonstrates it as a directed acyclic graph (DAG). This approach is distinctly different from a co-occurrence network, which can reparent only correlations between variables and is agnostic about their underlying causal relationships. For example, the *S. pneumoniae/S. aureus* module has direct effects on the microbial-mRNA metabolism module and the host neutrophil/IL-1 and type I IFN modules, which have a subsequent effect on the PPV use. Abbreviations: BCAA branched-chain amino acid, FDR false discovery rate, GPCR G-protein-coupled receptor, HR host response, IFN interferon, IL interleukin, NADH nicotinamide adenine dinucleotide hydrogen, MC microbial composition, PPV positive pressure ventilation.

were depleted[42], suggesting protective effects of Tregs in this population. Lastly, these potential mechanisms linking respiratory viruses, host immune response, airway microbiome, and bronchiolitis pathobiology are not mutually exclusive.

Using the metatranscriptome data, the current study also identified unique microbial functions−e.g., BCAA metabolism, oxidative stress response, NADH pathways−that are individually and/or synergistically related to the disease severity. First, research has shown that the lack of BCAAs (e.g., isoleucine)−essential nutrients in bacteria[43]− biosynthesis in *S. pneumoniae* lead to decreased growth, colonization, and expression of virulence factors[44]. Second, studies have also shown the role of oxidative stress response in the virulence of microbes in the oxygen-rich environment, such as the airway[45]. For example, *S. pneumoniae* employs predominantly enzymatic mechanisms (e.g., NADH oxidase, superoxide dismutase) to eliminate the effects of oxidative stress[45]. Indeed, loss of the NADH oxidase activity encoded by *nox* results in a decrease in the virulence of *S. pneumoniae*[46]. Additionally, NADH oxidase contributes to the virulence of *S. pneumoniae* as an adhesin−an important cell-surface component in the infectious process−and elicits a protective immune response in mice[47]. Lastly, research has also shown that direct interactions between RSV and *S. pneumoniae* alter microbial gene expression (e.g., *ply*, *pbp1A*), thereby increasing the virulence and worsening disease severity[48]. Our inferences−in conjunction with the existent evidence−indicate a complex interplay between respiratory viruses, these microbial species, their function, and host response in the airway, and their integrated contribution to the bronchiolitis pathobiology. Our data should facilitate further investigations to disentangle the complex web and to determine the role of modulating microbiome (e.g., prebiotics and probiotics) in the treatment of severe bronchiolitis.

The current study has several potential limitations. First, the study did not have "healthy controls". Yet, the study objective was not to evaluate the role of transcriptome and metatranscriptome in the development of bronchiolitis but to investigate their relationship with the disease severity *within* infants with bronchiolitis. Second, bronchiolitis involves inflammation of both upper and lower airways, while our study is based on nasopharyngeal specimens. The use of upper airway specimens is preferable because lower airway sampling (e.g., bronchoscopy) would be invasive in these young infants. Studies have suggested that upper airway sampling possibly represents the lung transcriptome[49] and microbiome[50] profiles in children. In contrast, studies in adults have reported similar but distinct microbial communities between concurrently sampled upper and lower airway specimens[51–53]. Third, the current study did not have mechanistic experiments to validate the identified microbial functions. Fourth, our inferences may be biased due to the relationship between the timing of treatments, specimen collections, and PPV use despite that the specimens were collected within a short time period. Fifth, while this study derives well-calibrated hypotheses that facilitate future experiments, our inferences warrant external validation. Lastly, although the study sample consisted of a racially/ethnically and geographically diverse multicenter cohort, our inferences should be generalized cautiously beyond infants hospitalized for bronchiolitis. Nonetheless, our observations remain highly relevant for 110,000 US children hospitalized each year−a population with a substantial health burden[4].

In conclusion, by applying an integrated-omics approach to dual-transcriptome data from a multicenter prospective cohort of 244 infants with bronchiolitis, we demonstrated a complex interplay between host response, microbial composition, and its function, and their integrated relationship with the disease severity. For example, host-type I IFN, neutrophil/IL-1, T-cell regulation, *S. pneumoniae/S. aureus*, microbial-BCAA metabolism, oxidative stress response, and NADH modules were associated with the risk of PPV use. Our observations should facilitate further research into the interplay between respiratory viruses, airway host response, microbiome, and disease pathobiology. This will, in turn, advance the development of targeted therapeutic measures (e.g., modification of immune response, microbiome composition and function) and help clinicians manage this population with a large morbidity burden.

## Methods
### Ethical statements
With the exception of specimen collection, all study participants were evaluated and treated as usual and without regard to this observational study. Parent/legal guardians were approached about participating after the medical team had finished their assessments and stabilized the study participant. The institutional review board at each of the participating hospitals approved the study. Written informed consent was obtained from the parent or guardian.

### Study design, setting, and participants
We collected and managed data using REDCap 10.0.30 (Nashville, TN, USA) electronic data capture tools. We analyzed data from a multicenter prospective cohort study of infants hospitalized for

bronchiolitis—the 35th Multicenter Airway Research Collaboration (MARC-35) study[21]. MARC-35 is coordinated by the Emergency Medicine Network (EMNet, www.emnet-usa.org), an international research collaboration with 247 participating hospitals. Site investigators enrolled infants (age < 1 year) hospitalized with bronchiolitis at 17 sites across 14 U.S. states using a standardized protocol during three consecutive bronchiolitis seasons (from November 1 through April 30) during 2011–2014[28]. The diagnosis of bronchiolitis was made according to the American Academy of Pediatrics bronchiolitis guidelines, defined as the acute respiratory illness with a combination of rhinitis, cough, tachypnoea, wheezing, crackles, or retraction[54]. We excluded infants with a pre-existing heart and lung disease, immunodeficiency, immunosuppression, or gestational age of <32 weeks, history of previous bronchiolitis hospitalization, or those who were transferred to a participating hospital >24 h after initial hospitalization.

Of 1016 infants enrolled into the cohort, the current analysis investigated 244 infants who were randomly selected for the dual-transcriptome profiling (Supplementary Table 2 and Supplementary Fig. 1). While some of the cohort data were used in a previous study (e.g., microbiome taxonomy data)[30], the current analysis tested for a hypothesis by using additional clinical data (e.g., acute severity outcomes), expanded study sample (e.g., patients with non-RSV infection), and microbiome function data.

### Data collection and measurement of virus and dual-transcriptome (host transcriptome and metatranscriptome) profiling

Clinical data (patients' demographic characteristics, and family, environmental, and medical history, and details of the acute illness) were collected via structured interview and chart reviews[21]. All data were reviewed at the EMNet Coordinating Centre (Boston, MA, USA), and site investigators were queried about missing data and discrepancies identified by manual data checks. In addition to the clinical data, nasopharyngeal airway specimens were collected by trained site investigators using the standardized protocol that was utilized in a previous cohort study of children with bronchiolitis[21,55]. All sites used the same collection equipment (Medline Industries, Mundelein, IL, USA) and collected the specimens within 24 h of hospitalization. For the collection, the child was placed supine, 1 mL of normal saline was instilled into one naris, and mucus was removed by means of an 8 French suction catheter. This procedure was performed once on each nostril. After specimen collection from both nares, 2 mL of normal saline was suctioned through the catheter to clear the tubing and ensure that a standard volume of aspirate was obtained. Once collected, the nasopharyngeal aspirate specimen was added to the transport medium at a 1:1 ratio. The specimens were immediately placed on ice within 1 h of collection and then stored at −80 °C within 24 h of collection[21,55].

These specimens underwent (1) real-time reverse transcription PCR to test for 17 respiratory viruses (including RSV and RV) using real-time polymerase chain reaction (RT-PCR) assays (Supplementary Table 6) in the nasopharyngeal airway at Baylor College of Medicine (Houston, TX, USA) and (2) dual-transcriptome profiling through RNAseq at the University of Maryland (Baltimore, MD, USA).

### RNA extraction, RNA sequencing, and quality control

Total RNA was isolated from the nasopharyngeal specimens using Trizol LS reagent (ThermoFisher Scientific, Waltham, MA, USA) in combination with the Direct-zol RNA Miniprep Kit (Zymo Research, Irvine, CA, USA). RNA quantity was measured with the Qubit 2.0 fluorometer (ThermoFisher Scientific, Waltham, MA, USA); its quality was assessed with the Agilent Bioanalyzer 2100 (Agilent, Palo Alto, CA, USA) using the RNA 6000 Nano kit. Total RNA underwent DNase treatment using the TURBO DNA-free™ Kit (ThermoFisher Scientific, Waltham, MA, USA) and rRNA reduction for both human and bacterial

rRNA using NEBNext rRNA Depletion Kits (New England Biolabs, Ipswich, MA, USA). RNA was prepared for sequencing using the NEBNext Ultra II Directional RNA Library Prep Kit (New England Biolabs, Ipswich, MA, USA) and sequenced on an Illumina NovaSeq6000 using an S4 100PE Flowcell (Illumina, San Diego, CA, USA). All RNAseq samples had sufficient sequence depth (mean, 8,067,019 pair-end reads/sample) to obtain a high degree of sequence coverage.

### Nasopharyngeal airway host transcriptome

Transcript abundances from clean RNAseq reads were estimated in Salmon using the human transcriptome (hg38) and the mapping-based mode[56]. We first generated a decoy-aware transcriptome and then quantified the reads using Salmon's default settings and the following flags: –validateMappings, –recoverOrphans, –seqBias, and –gcBias. Salmon is fast and accurate, corrects for potential changes in gene length across samples (e.g., from differential isoform usage), and has great sensitivity.

### Nasopharyngeal airway microbial composition and function profiling

Raw sequence reads were filtered and trimmed for adapters and contaminants using the k-mers strategy in KneadData v0.10.0[57]. We used PathoScope 2.0[58] and the expanded Human Oral Microbiome Database (eHOMD) database[59] to infer bacterial taxonomy. This database only includes bacteria, hence viruses and fungi were classified using Kraken[60] and the maxikraken2_1903 database (https://lomanlab.github.io/mockcommunity/mc_databases.html). Samples with <1000 reads, singletons, and strains not present in at least 10% of the samples were eliminated. The metatranscriptomic analysis obtained 1,968,352,599 merged sequences and identified 320 microbial species after singleton removal.

We inferred microbial gene functions and Gene Ontologies from the metatranscriptomic contigs annotated with EggNOG-mapper[61,62]. Briefly, we removed the reads of human origin by mapping against the human genome sequence using Bowtie2[63]. Then, we collected all the unassigned reads using the MEGAHIT algorithm[64], after gene annotation, we assigned the reads to contigs using the HISAT2 aligner[65,66], as the last step to count the transcript we used HTSeq[67].

### Outcome measures

The primary outcome was higher disease severity defined by the use of PPV (continuous positive airway pressure and/or intubation with mechanical ventilation) during the hospitalization for bronchiolitis[20]. The secondary outcome was intensive care use defined by the use of PPV and/or intensive care unit admission during the hospitalization for bronchiolitis[21]. We used PPV use as the primary outcome as it is considered more specific than intensive care use[68].

### Statistical analyses

In the current study, our aims are to investigate (1) the individual relationship of nasopharyngeal airway dual-transcriptome—(i) host response (transcriptome), (ii) microbial composition, and (iii) microbial function (metatranscriptome)—with disease severity and (2) their integrated relationships. The analytic workflow is summarized in Fig. 1.

We examined the association of each omics data element with the risk of PPV use at the individual data level. First, in the examination of the host transcriptome data, we conducted differential expression gene and functional pathway analyses by comparing infants with PPV use to those without PPV use. To investigate whether genes for specific biological pathways are enriched, we conducted a functional class scoring analysis using R clusterProfiler and fgsea packages[69–71]. Second, in the nasopharyngeal microbial composition data, we investigated the relationship of the abundance of the top 20 most abundant microbial species with the PPV outcome by computing the $\log_2$ fold change of median abundance. Third, in the microbial function data, we

conducted differential expression gene and functional pathway analyses, similar to the analysis of the host transcriptome data.

Next, to reduce the dimensionality of the host transcript, microbial composition, and microbial function data, and to identify co-expression networks (modules)—that is, clusters of densely interconnected genes or species—we applied a WGCNA approach by using R *wgcna* package[29] As low-expressed or non-varying genes represent noise in WGCNA[29], we selected differentially enriched transcripts and metatranscripts with an FDR of <0.40 and high variance (top 90%) and microbial species with high variance for the WGCNA. We identified a soft thresholding power for network construction and confirmed the whole-network connectivity distribution by log-log plots (Supplementary Fig. 9). We then merged highly correlated modules using a cut height that is chosen to identify an optimal number of adequately sized modules for the analysis[29,72]. To identify biologically meaningful pathways within each of the transcriptome and metatranscriptome modules, we performed functional pathway analyses (gene ontology enrichment analyses) using R *clusterProfiler* package[70,71].

We investigated the integrated associations of these dual-transcriptome modules with each severity outcome by constructing a logistic regression model with ridge regularization[73] that adjusts for potential confounders (sex, age, and respiratory viruses [RSV, RV, and coinfection]). Ridge regularization is a statistical approach that mitigates overfitting in the setting of a limited sample size[73]. We used leave-one-out cross-validation to yield an optimal regularization parameter that minimizes the sum of least squares plus a shrinkage penalty by using R *glmnet* and *caret* packages[74,75]. We also estimated 95% CI by a bootstrap method with 2000 replicates. Lastly, to visualize relationship between major clinical characteristics and dual-transcriptome modules, we developed a co-occurrence plot based on the Spearman's correlation by using Cytoscape[76]. Additionally, to identify the underlying causal relationships between the dual-transcriptome modules and PPV use, we utilized the PC algorithm implemented in R *pcalg* package[77]. This causal structure learning approach recovers the underlying causal pathways through the conditional independence relationships in the empirical data. In the sensitivity analysis, we repeated the integrated analysis for the intensive care use outcome. We also constructed the integrated models adjusting for race/ethnicity in addition to age, sex, and virus. We reported all $P$ values as two-tailed, with $P < 0.05$ considered statistically significant. To account for multiple comparisons, we used the Benjamini–Hochberg FDR method, as appropriate[78]. We analyzed the data with the use of R version 3.6.1 (R Foundation, Vienna, Austria).

### Reporting summary
Further information on research design is available in the Nature Research Reporting Summary linked to this article.

## Data availability
The data that support the findings of this study are available on the NIH/NIAID ImmPort (https://www.immport.org/shared/study/SDY1883) through controlled access to be compliant with the informed consent forms of MARC-35 study and the genomic data sharing plan. Source data without participant-level data are provided with this paper as a Source Data file. Source data are provided with this paper.

## Code availability
Computational code from the study is available at https://zenodo.org/record/6590728#.Yt53LXbMJdg.

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

## Acknowledgements

This study was supported by grants from the National Institutes of Health (Bethesda, MD): U01 AI-087881, R01 AI-114552, R01 AI-108588, R01 AI-134940, and UG3/UH3 OD-023253. M.P.-L. was partially supported by the Margaret Q. Landenberger Research Foundation, the NIH National Center for Advancing Translational Sciences (Award Number UL1TR001876), and the Fundação para a Ciência e a Tegnologia (T495756868-00032862). The content of this manuscript is solely the responsibility of the authors and does not necessarily represent the official views of the National Institutes of Health. The funding organizations were not involved in the collection, management, or analysis of the data; preparation or approval of the manuscript; or decision to submit the manuscript for publication. We thank the MARC-35 study hospitals and research personnel for their ongoing dedication to bronchiolitis and asthma research (Supplementary Table 1), and Ashley F. Sullivan, MS, MPH and Janice A. Espinola, MPH (Massachusetts General Hospital, Boston, MA) for their many contributions to the MARC-35 study. We also thank Alkis Togias, MD, at the National Institutes of Health (Bethesda, MD) for helpful comments about the study results.

## Author contributions

M.F. carried out the main statistical analysis, drafted the initial manuscript, and approved the final manuscript as submitted. Y.R. assisted statistical analysis, reviewed the manuscript, and approved the final manuscript. M.P.-L. conducted microbiome and transcriptome analyses, carried out statistical analysis, reviewed the manuscript, and approved the final manuscript. R.J.F. conducted specimen processing, supervised RNA sequencing and data generation, reviewed and revised the manuscript, and approved the final manuscript as submitted. J.C.C. and J.M.M. collected the data, reviewed and revised the manuscript, and approved the final manuscript as submitted. P.A.P conducted virus testing and interpreted the results, reviewed and revised the manuscript, and approved the final manuscript as submitted. Z.Z. assisted statistical analysis, reviewed the manuscript, and approved the final manuscript. C.A.C. conceptualized and designed the study, obtained funding, collected the data, supervised the conduct of study and the analysis, critically reviewed and revised the initial manuscript, and approved the final manuscript as submitted. K.H. conceptualized the study, obtained funding, supervised the statistical analysis, reviewed and revised the initial manuscript, and approved the final manuscript as submitted.

## Competing interests

The authors declare no competing interests.
