## [Peer Review File · Nature Communications]

Integrated relationship of nasopharyngeal airway host
response and microbiome
associates with bronchiolitis severityREVIEWER COMMENTS

Reviewer #1 (Remarks to the Author):

This study reports results from dual-transcriptomic analysis of nasopharyngeal airway samples acquired from infants with bronchiolitis, in which the investigators sought to determine microbial and host response expression patterns associated with more severe bronchiolitis, defined as use of positive-pressure ventilation. Virus-related severe bronchiolitis is an important medical illness in infants and has been associated with risk for later lung disease, in particular asthma. The investigators here leverage data generated from NP samples procured from a large consortium study of infant bronchiolitis. The ability to have acquired such samples and generated molecular data from a difficult biological niche (nasopharyngeal space in infants) is impressive. There is increasing interest in augmenting discovery-driven studies of the microbiome by melding techniques to glean insight into potential functional interactions at play. Major comments/concerns and queries are:

1. First, it is somewhat unclear if some of the data used in this study have been previously analyzed and published. Information from another recent published paper from this group invites the question (Raita et al. Nat Comm 2021). Although Raita et al. analyzed NP molecular data in relation to a different clinical outcome (childhood asthma), there are very similar if not identical statements in the Methods. For example, between the two papers, the same mean number of paired-end reads/sample from the RNAseq is stated (mean=8,067,019). Likewise, in Raita et al., it is stated “the metatranscriptomic analysis obtained 1,968,352,599 merged sequences and identified 323 microbial lineages after singleton removal”. This is nearly identical statement to that included in this manuscript, except for mention of 320 rather than 323 microbial lineages. So, it would appear that these data, including metatranscriptomic analysis, may have been previously generated and conducted. If this is the case, but being analyzed by different computational approaches, this should be made clearer as to what is or isn't redundant.
2. It is stated that a random number of subjects were selected for inclusion in this study (similar statement in earlier paper also). Although the overall cohort was quite large, was there any overlap in the subjects included in this study vs. prior studies from the same parent cohort (and thus, whether some of the same data may have been reanalyzed)? Moreover, it is unclear what the process was to ensure “randomly selected” subjects for inclusion.
3. The number(%) of subjects in the cohort who required PPV is quite low, which seems a little surprising. Does this conform with clinically observed rates of PPV need in infant bronchiolitis? The % requiring PPV appears to be similar between the analyzed vs. non-analyzed cohort. So, as follow up to the prior question (#2), how were the investigators able to ensure that a reasonable number of subjects who required PPV were selected for inclusion? Moreover, with a case rate of essentially 7% (18/244) for the primary outcome, how generalizable are the findings to other cases/cohorts of infants with bronchiolitis? It appears that internal cross-validation was done with the integrated dual-transcriptome module analysis, but there was no external validation using an independent cohort, which would be more ideal and potentially informative.
4. The methods state that specimens were collected within 24 hours of hospitalization. In the integrated analysis of dual-transcriptome modules, it is stated that the ridge regression models were adjusted for age, sex and respiratory viruses detected. Did the authors consider additional covariates such as race, treatments administered up to time of collection,

and time to sample collection after hospital admission? Table 1 indicates that 42% of the cohort was white, 23% black, 31% Hispanic. What was the racial distribution among the cases (the 18 subjects who required PPV)? As clinical outcomes might vary by race (and racial disparities in lung disease outcomes is well recognized). Also, though 24 hours is a relatively short period, nonetheless it is possible that some gene expressions change greatly within the first 6-8 hours, for example, compared to 16-24 hours later. This could be and is particularly true for bacteria.

5. A specific description could not be found of what exactly the nasopharyngeal airway sample collected for analysis was. Prior studies are referenced (and took this reviewer a bit of searching to locate the information), but it would be appropriate to restate. Especially since NP sampling for microbiome-related investigations has varied widely across studies (swabs? lavage/washes? aspirates?). Moreover, for reporting consistency, this would conform with recently published STORMS guidelines.

6. Lines 168-169 state that “studies have shown that upper airway sampling provides a reliable representation of the transcriptome and lung microbiome”. While the former is likely true, it should be noted that the latter (lung microbiome) is likely only true in children or infants. The referenced studies are in children, as applicable to this study, but it should be known/noted that studies in adults, have shown that the nasal/nasopharyngeal microbiome is quite distinct in bacterial composition from the lower airways/lungs (when latter is sampled directly for paired comparisons). So the current statement should be revised to reflect this caveat so that the context is clear.

7. Some of the specific host transcriptome and microbial compositional features found to be associated with more severe bronchiolitis are somewhat expected based on prior evidence (e.g. neutrophil pathways, downregulation of type 1 IFN responses.). Similar with the microbiota compositional results (*S. pneumoniae*). So these are confirmatory of the larger literature which is good but also not novel. Although they used dual RNAseq approach, it is not reported whether transcripts from other types of microbes were found, for example from fungi or other viruses. Even if these were not found to be associated with their outcomes of interest, it would be of interest to comment on whether or not such transcripts were even observed in the transcriptomic data. As these are such understudied elements/members of respiratory microbiota. Moreover, given the strong association with *S.pneumoniae*/*S.aureus* module, what does this mean for potential clinical management of bronchiolitis? Are these bacterial co-pathogens with RSV that should be considered for targeted treatment with anti-bacterial agents? Of course a clinical recommendation cannot be made from this study nor from similar lines of evidence from other groups, but it does invite the question and a possible comment in the discussion.

8. Given the above comment, it would have been nice to see more Discussion about how the current findings compare/contrast with prior results from this group and other cohorts. There is some discussion of this, but it feels limited. For example, in light of their earlier paper that reported different integrated endotypes of RSV infection (and risk for asthma). I acknowledge that the focus of the current manuscript is on an entirely different clinical outcome, but nonetheless it seems there may be potential overlap in results of interest for a comparative discussion. The mentioned paper is just one suggestion since RSV was the predominant virus found in the analyzed cohort here (and severe bronchiolitis in turn is associated with increased risk for asthma). But there might be relevant results from other studies that could also be considered for a more expanded Discussion. Similarly, it would be nice to see more Discussion on the microbial functional expressions identified, including

mention of whether any of the functions identified could be mapped to specific bacteria. There is some such discussion (lines 148-160), however no mechanistic experiments were performed to potentially validate the computational findings. To determine, for example, if specific genes in the reported pathways were indeed up-regulated or down-regulated in specific bacteria by targeted PCR.

9. There is a typo in Figure 1 related to misspelling of transcriptome.

Reviewer #2 (Remarks to the Author):

I am reviewing a manuscript by Fujiogi et al. titled "Integrated relationship of nasopharyngeal airway 1 host response and microbiome with bronchiolitis severity: dual-transcriptomic profiling" I found the manuscript excellently constructed and the conclusions sound. I have a few, small recommendations and two particular criticisms. If code/data was available for review, it would greatly strengthen my confidence in the authors' results.

1. Some of the figures need to have labels, especially the volcano plots. Some of the integrated data can be used in co-occurrence plots, I think some of the data can be plotted using cytoscape and could be a pretty good figure regarding PPV outcomes.

2. I would say that nasopharyngeal microbiome is not a lower airway microbiome surrogate, there are some publications that make reference that upper airway microbiome is equivalent to the lower airway microbiome, but generally this is not a widely accepted view and still controversial. I would suggest that the authors change the sentence in strengths/weakness section.

3. How did the authors take into account time to PPV use? This was probably the most clinically oriented question I had that did not get addressed in the data or results. Was there any lead time or length time bias involved with how / when the subjects required PPV? I wonder if tit would strengthen their observations or transcriptomic signatures by taking into account time to PPV?

4. Lack of code/data for review is the manuscripts greatest weakness. For a largely bioinformatic manuscript, the code should be available for review. While I can understand that raw fastq data is not available, efforts should be made for the microbiota data to be available on NIH SRA. I would recommend that data objects be generated and uploaded to a repository such as github if raw data will be going to dGAP. The data objects can contain the annotated gene/transcriptome/metatranscriptome without the raw data. Code should contain enough information to be able to generate the figure from the data objects.

(<https://www.springernature.com/gp/authors/research-data-policy/data-availability-statements/12330880>) (Mirzayi et al. Nature Medicine 2021) (Langille et al. Microbiome 2018).

I will be using the page PDF page numbers and the leftmost numbers

Abstract

No recommendations

Introduction

Page 5, Line 22

Figure 1 is an excellent figure and certainly signposts the goals of the manuscript. I wanted to thank the authors for including a figure to describe their analysis and conclusions.

Results

Page 7, Line 29

Figure 2A please apply the labels to the R generated figure – I believe it should can be layered in. Also, I would suggest to label the directions (e.g., upregulated in PPV use vs. upregulated in subjects w/o PPV use).

Page 8, Line 42

Figure 3A This is good figure, but the phylum legend has mismatched colors, it's a bit distracting. I

Page 8, Line 39

I also found it interesting that the presence of one bacterial species would have divergent results, such as having Haemophilis as an increase of LogFold change as a predictor for PPV use, and lower Logfold change associated with intensive care use? Were the criteria different? Another taxa is Staph epi and pneumococcus which also shows differences between LogFold change in PPV use and intensive care use. I would have expected that increase in LogFold Change in these taxa predicted negative results.

Page 8, Line 52

Figure 4A would benefit from the same recommendations re: Figure 2A.

Page 9, Line 60-67

I am wondering if the investigators could demonstrate the correlations with something like cytoscape to show correlations between microbes, microbial functions, pathways, clinical factors, and outcomes. It would be a nice way to summarize otherwise busy tables. The information is present, but

Discussion:

Page 11, Line 105

“To the best of our knowledge, ...” I think this is a particular strength of the manuscript with this excellent multi-omic evaluation of a sample cohort of pediatric subjects with and without PPV ventilation. One of my particular interests or next steps that the authors could do is to use the observations in a validation cohort? I wonder if this would make the manuscript stronger if there was another cohort which the authors could test their observations and predict those subjects who may go onto require PPV.

Page 14, Line 163

The strengths and weaknesses of the manuscript are well thought out. I am particularly impressed with the argument for nasopharyngeal testing. I would however cautiously agree with nasopharyngeal testing is equivalent to lower airway testing "...upper airway sampling provides a reliable representation ..." (line 168) I think simply stating that NP testing can predict some understanding of the lung microbiome/transcriptome without drawing equivalence would probably be a better way of stating why we should consider NP testing vs. lung testing.

Methods:

Page 16, Line 207

What mechanism of random selection was used to choose the infants for -omic profiling?

Page 17, Line 22

Timing of respiratory failure? I am not sure if this was clearly described, but when did the subjects require PPV ? Did timing of respiratory failure (requiring PPV) impact the multi-omic results? How was time to respiratory failure accounted for in the data/results?

Page 21, Line 317 and 320

Data that does not include host (human data) such as the microbiome/metatranscriptome should be uploaded to the NIH SRA. If there is concern, the authors could filter any possible human contaminant DNA/RNA and upload that data to the NIH SRA. If the reviewers are concerned re: privacy the SRA can generate a reviewer link for review (thus private). This has not been an issue with other publications. If possible, could the authors provide more information regarding the controlled access? (Will they be uploading the data through dGAP?)

Code availability, it is unacceptable that the code is unavailable for review. While understandable, there is no suggestion that any of the code is proprietary, thus it should be shared for review. Given that a significant amount of their work is bioinformatics/code this should be uploaded to a code repository. The data objects generated in R could be uploaded into github (this could avoid issues with raw fastq data) and code to generate the figures.

REVIEWER COMMENTS

COMMENTS FROM REVIEWER #1:

This study reports results from dual-transcriptomic analysis of nasopharyngeal airway samples acquired from infants with bronchiolitis, in which the investigators sought to determine microbial and host response expression patterns associated with more severe bronchiolitis, defined as use of positive-pressure ventilation. Virus-related severe bronchiolitis is an important medical illness in infants and has been associated with risk for later lung disease, in particular asthma. The investigators here leverage data generated from NP samples procured from a large consortium study of infant bronchiolitis. The ability to have acquired such samples and generated molecular data from a difficult biological niche (nasopharyngeal space in infants) is impressive. There is increasing interest in augmenting discovery-driven studies of the microbiome by melding techniques to glean insight into potential functional interactions at play. Major comments/concerns and queries are:

[Response]

We thank the reviewers for these positive comments.

1. First, it is somewhat unclear if some of the data used in this study have been previously analyzed and published. Information from another recent published paper from this group invites the question (Raita et al. Nat Comm 2021). Although Raita et al. analyzed NP molecular data in relation to a different clinical outcome (childhood asthma), there are very similar if not identical statements in the Methods. For example, between the two papers, the same mean number of paired-end reads/sample from the RNAseq is stated (mean=8,067,019). Likewise, in Raita et al., it is stated “the metatranscriptomic analysis obtained 1,968,352,599 merged sequences and identified 323 microbial lineages after singleton removal”. This is nearly identical statement to that included in this manuscript, except for mention of 320 rather than 323 microbial lineages. So, it would appear that these data, including metatranscriptomic analysis, may have been previously generated and conducted. If this is the case, but being analyzed by different computational approaches, this should be made clearer as to what is or isn’t redundant.

[Response]

We appreciate the opportunity to clarify these important points. First, as for the data source, the current study has used a combination of *existent* cohort data (e.g., the microbiome taxonomy data) and *newly*-generated data in the 35th Multicenter Airway Research Collaboration (MARC-35) cohort. MARC-35 is an NIH-funded large 17-center prospective cohort of infants hospitalized with bronchiolitis with comprehensive clinical phenotyping and >6-year follow-up. Publishing a new and distinct hypothesis by leveraging large cohort data—as done in similarly large cohort studies (e.g., the Framingham Heart

Study¹, UK Biobank²)—is cost-effective and encouraged. For example, we have previously published a distinctly different paper in *Nat Commun* (Raita et al. 2021;12(1):3601)³ based on the clinical (e.g., asthma outcomes) and taxonomy data of MARC-35—some of which have been used in this manuscript.

In the current study, we have added three new data components:

- 1) The current study has used microbiome *function* data, which are newly generated by applying novel bioinformatic approaches to meta-transcriptome data. The data have enabled for us to test, for the first time, the complex interplay between the airway host response, microbiome (both composition and function), and their integrated contributions to disease severity of bronchiolitis.
- 2) The current study has also *expanded the study sample* to include non-respiratory syncytial virus (RSV). Although the previous *Nat Commun* article used only the data of RSV infection, bronchiolitis is also caused by a multitude of virus pathogens (e.g., rhinovirus). The present paper has more-comprehensively examined the bronchiolitis population.
- 3) Based on *a priori*-defined new hypothesis, the current study has investigated *the acute severity outcomes* (e.g., the use of positive pressure ventilation), which were not examined in the previous paper.

In summary, the current study extends the previous MARC-35 articles by using newly-added data (i.e., microbial function, an expanded study sample including non-RSV infection, and acute severity outcomes) and testing a new hypothesis (i.e., the integrated relationship of airway host response, microbiome composition, and function influences the disease severity).

As requested, we have clarified this point in the Methods section as follows: “While some of the cohort data were used in a previous study (e.g., microbiome taxonomy data)³⁰, the current analysis tested for a novel hypothesis by using additional clinical data (e.g., acute severity outcomes), expanded study sample (e.g., patients with non-RSV infection), and microbiome function data.” (page 18, para 1).

Second, as for the number of microbial lineages, the current study has used and reported the *species*-level data, while the previous study³⁵ described the number of microbial *strains*. As suggested, we have clarified this point in the Methods section as follows: “The metatranscriptomic analysis obtained 1,968,352,599 merged sequences and identified 320 microbial species after singleton removal.” (page 20, para 2).

2. It is stated that a random number of subjects were selected for inclusion in this study (similar statement in earlier paper also). Although the overall cohort was quite large, was there any overlap in the subjects included in this study vs. prior studies from the same parent cohort (and thus, whether some of the same data may have been reanalyzed)? Moreover, it is unclear what the process was to ensure “randomly selected” subjects for inclusion.

[Response]

We appreciate the opportunity to clarify this point. The current study used data from 244 infants who were randomly selected from the MARC-35 overall cohort (n=1016) (as shown below). To clarify this point, we have shown the study flow diagram in **Supplemental Figure 1** and cited it in the Results (page 7, para 1) and Methods (page 18, para 1) sections.

As clarified above, there is some overlap in the study sample between the previous *Nat Commun* study³ and the current study. While the former was limited to only those with RSV infection, the current study expanded the study sample to include patients with non-RSV infection. The present paper has more-comprehensively examined the bronchiolitis population.

Supplemental figure 1. Study flow diagram

The differences in the analytic and non-analytic cohorts are summarized in **Table E1**.

* The transcriptome and metatranscriptome data are obtained in 244 infants who were *randomly* selected from the overall cohort

3. The number(%) of subjects in the cohort who required PPV is quite low, which seems a little surprising. Does this conform with clinically observed rates of PPV need in infant bronchiolitis? The % requiring PPV appears to be similar between the analyzed vs. non-analyzed cohort. So, as follow up to the prior question (#2), how were the investigators able to ensure that a reasonable number of subjects who required PPV were selected for inclusion? Moreover, with a case rate of essentially 7% (18/244) for the primary outcome, how generalizable are the findings to other cases/cohorts of infants with bronchiolitis? It appears that internal cross-validation was done with the integrated dual-transcriptome module analysis, but there was no external validation using an independent cohort, which would be more ideal and potentially informative.

[Response]

A previous study of nationally-representative data of infants hospitalized for bronchiolitis (n=490,650) has demonstrated a similar rate of positive pressure ventilation (PPV)⁴. Indeed, the proportion of children who underwent PPV was 5 % in the U.S. in 2016. Another prospective cohort study (n=2,207) has also shown that PPV use was 7% in U.S. children hospitalized for bronchiolitis⁵. Our finding (7%) was similar to these data. Additionally, 244 infants in this study were *randomly* selected from the overall cohort with no significant difference in the rate between the analytic and non-analytical cohorts, arguing against selection bias.

As for the generalizability of the inference, we agree with the reviewer. Accordingly, we have acknowledged this point in the Limitations section. The text now states: "...while this study derives novel and well-calibrated hypotheses that facilitate future experiments, our inferences warrant external validation." (page 16 para 2).

4. The methods state that specimens were collected within 24 hours of hospitalization. In the integrated analysis of dual-transcriptome modules, it is stated that the ridge regression models were adjusted for age, sex and respiratory viruses detected. Did the authors consider additional covariates such as race, treatments administered up to time of collection, and time to sample collection after hospital admission? Table 1 indicates that 42% of the cohort was white, 23% black, 31% Hispanic. What was the racial distribution among the cases (the 18 subjects who required PPV)? As clinical outcomes might vary by race (and racial disparities in lung disease outcomes is well recognized). Also, though 24 hours is a relatively short period, nonetheless it is possible that some gene expressions change greatly within the first 6-8 hours, for example, compared to 16-24 hours later. This could be and is particularly true for bacteria.

[Response]

We thank the reviewer for these helpful suggestions. As suggested by the reviewer, we have evaluated the association between race/ethnicity and PPV use and performed sensitivity analyses that also adjust for race/ethnicity as a potential confounder. First, there was no significant association between race/ethnicity and PPV use (P=0.59; please see **Table** below). Second, in the sensitivity analysis adjusting for race/ethnicity (in addition to age, sex, and virus), the results for the PPV risk did not materially change (**Supplemental Figure 6**; next page). For example, the odds ratio (95% confidence interval [CI]) for T cell regulation module was 0.24 (0.11-0.53) in the main analysis and 0.24 (0.11-0.53) in the sensitivity analysis. Likewise, in the intensive care use model, the additional adjustment for race/ethnicity did not change the results materially (**Supplemental Figure 7**; next page).

As suggested, we have added these analyses to the Results and Methods sections. The text now states: “Additionally, in the sensitivity analysis adjusting for race/ethnicity (in addition to age, sex, and virus), the results did not materially change (**Supplemental Figures 6-7**).” (the Results section, page 10, para 1) and “In the sensitivity analysis, we repeated the integrated analysis for the intensive care use outcome. We also constructed the integrated models adjusting for race/ethnicity in addition to age, sex, and virus.” (the Methods section, page 23, para 1).

Table (Only for review). Association of race/ethnicity with the risk of positive pressure ventilation use in infants hospitalized for bronchiolitis

	No PPV use (n=226)	PPV use (n=18)	P-value
Race/ethnicity			0.59
Non-Hispanic white	92 (90%)	10 (10%)	
Non-Hispanic black	54 (95%)	3 (5%)	
Hispanic	72 (95%)	4 (5%)	
Others	8 (89%)	1 (11%)	

Abbreviation: PPV, positive pressure ventilation

Supplemental Figure 6. Sensitivity analysis adjusting for age, sex, race/ethnicity, and virus: Integrated relationship of the dual-transcriptome modules with the risk of positive pressure ventilation use in infants hospitalized for bronchiolitis

Supplemental Figure 7. Sensitivity analysis adjusting for age, sex, race/ethnicity, and virus: Integrated relationship of the dual-transcriptome modules with the risk of intensive care use in infants hospitalized for bronchiolitis

The adjusted odds ratio for the outcome was estimated per one unit increased in the eigen-value (the first principal component) of the corresponding module by fitting a multivariable logistic regression model with ridge regularization. In the model, we adjusted for age, sex, race/ethnicity, and respiratory virus. Statistically significant modules are in **bold**.

Abbreviations: BCAA, branched-chain amino acid; FDR, false discovery rate; GPCR, G-protein-coupled receptor; HR, host response; IFN, interferon; IL, interleukin; NADH, nicotinamide adenine dinucleotide hydrogen; MC, microbial composition

Furthermore, we note that we do not have the exact information on the exact timing of specimen collection within the 24-hour interval. As suggested, we have acknowledged this point in the Limitations section. The text now states: “Third, our inferences may be biased due to the relationship between the timing of treatments, specimen collections, and PPV use...” (page 16, para 1).

5. A specific description could not be found of what exactly the nasopharyngeal airway sample collected for analysis was. Prior studies are referenced (and took this reviewer a bit of searching to locate the information), but it would be appropriate to restate. Especially since NP sampling for microbiome-related investigations has varied widely across studies (swabs? lavage/washes? aspirates?). Moreover, for reporting consistency, this would conform with recently published STORMS guidelines.

[Response]

We thank the reviewer for this suggestion. The study used nasopharyngeal aspirate specimens. As suggested, we have confirmed the STORMS guideline and added a detailed description of the collection methods to the Methods section as follows: “For the collection, the child was placed supine, 1 mL of normal saline was instilled into one naris, and mucus was removed by means of an 8 French suction catheter. This procedure was performed once on each nostril. After specimen collection from both nares, 2 mL of normal saline was suctioned through the catheter to clear the tubing and ensure that a standard volume of aspirate was obtained. Once collected, the nasopharyngeal aspirate specimen was added to the transport medium at a 1:1 ratio. The specimens were immediately placed on ice within 1 hour of collection and then stored at -80°C within 24 hours of collection.” (page 18, para 3).

6. Lines 168-169 state that “studies have shown that upper airway sampling provides a reliable representation of the transcriptome and lung microbiome”. While the former is likely true, it should be noted that the latter (lung microbiome) is likely only true in children or infants. The referenced studies are in children, as applicable to this study, but it should be known/noted that studies in adults, have shown that the nasal/nasopharyngeal microbiome is quite distinct in bacterial composition from the lower airways/lungs (when latter is sampled directly for paired comparisons). So the current statement should be revised to reflect this caveat so that the context is clear.

[Response]

As suggested, we have highlighted this point in the Limitations section as follows: “our study is based on nasopharyngeal specimens. The use of upper airway specimens is preferable because lower airway sampling (e.g., bronchoscopy) would be invasive in these young infants. Studies have suggested that upper airway sampling possibly represents the lung transcriptome⁴⁹ and microbiome⁵⁰ profiles in children. In contrast, studies in adults have reported similar but distinct microbial communities between concurrently sampled upper and lower airway specimens⁵¹⁻⁵³.” (page 15, para 2 to page 16, para 1).

7. Some of the specific host transcriptome and microbial compositional features found to be associated with more severe bronchiolitis are somewhat expected based on prior evidence (e.g. neutrophil pathways, downregulation of type 1 IFN responses.). Similar with the microbiota compositional results (*S. pneumoniae*). So these are confirmatory of the larger literature which is good but also not novel. Although they used dual RNAseq approach, it is not reported whether transcripts from other types of microbes were found, for example from fungi or other viruses. Even if these were not found to be associated with their outcomes of interest, it would be of interest to comment on whether or not such transcripts were even observed in the transcriptomic data. As these are such understudied elements/members of respiratory microbiota. Moreover, given the strong association with *S.pneumoniae*/*S.aureus* module, what does this mean for potential clinical management of bronchiolitis? Are these bacterial co-pathogens with RSV that should be considered for targeted treatment with anti-bacterial agents? Of course a clinical recommendation cannot be made from this study nor from similar lines of evidence from other groups, but it does invite the question and a possible comment in the discussion.

[Response]

We thank the reviewer for these insightful comments. First, as requested, we have annotated fungi and viruses using the metatranscriptome data. We have identified a total of 340 fungal species in the nasopharyngeal airway of these infants with bronchiolitis. We have shown the relationship of the 10 most abundant fungal species with the severity outcome in **Supplemental Figure 3** (next page). Of these, 9 species were significantly associated with the PPV risk (FDR<0.001). For example, a higher abundance of *Malassezia restricta* was significantly associated with a higher PPV risk (FDR<0.001). In addition, we have identified a total of 111 viruses (please see **Table**; page 13-14 of this letter). We have shown the relationship of the 10 most abundant viruses with the severity outcome in **Figure** (page 15 of the letter), while this does not add substantially to the existent PCR data on 17 respiratory viruses. As requested, we have summarized these to the text as follows: “Additionally, a total of 340 fungal species were detected. Of 10 most abundant species, 9 species were significantly associated with the PPV risk (FDR<0.001; **Supplemental Figure 3**). For example, a higher abundance of *Malassezia restricta* was significantly associated with a higher PPV risk (FDR<0.001).” (the Results section, page 8, para 2) and “Raw sequence reads were filtered and trimmed for adapters and contaminants using the k-mers strategy in KneadData v0.10.0⁵⁷. We used PathoScope 2.0⁵⁸ and the expanded Human Oral Microbiome Database (eHOMD) database⁵⁹ to infer bacterial taxonomy. This database only includes bacteria, hence viruses and fungi were classified using Kraken⁶⁰ and the maxikraken2_1903 database (https://lomanlab.github.io/mockcommunity/mc_databases.html).” (the Methods section, page 20, para 2)

Supplemental Figure 3. Relationship of abundant fungal species with the risk of higher severity in infants hospitalized for bronchiolitis

The pirate plots show the comparison of the distribution of annotated ten most abundant fungal species in the nasopharyngeal airway of infants hospitalized for bronchiolitis, according to the PPV use. Each point represents each infant. The grey bar and rectangle represent the mean and 95% confidence interval. In the violin plots, the width represents the probability that infants take on a specific relative abundance. The between-group differences in the abundance were tested by fitting Poisson regression models. Abbreviations: FDR, false discovery rate; PPV, positive pressure ventilation

Table. Detected viruses using nasopharyngeal metatranscriptome data in infants hospitalized for bronchiolitis

Detected viruses by metatranscriptomic profiling		
Actinomyces virus Av1	Pestivirus sp1	Propionibacterium virus Stormborn
Begomovirus sp1	Picornavirales sp1	Propionibacterium virus Wizzo
Bell pepper alphaendornavirus	Pneumoviridae sp1	Pseudomonas phage Lu11
Betacoronavirus 1	Porcine type-C oncovirus	Pseudomonas virus KPP25
Bovine orthopneumovirus	Propionibacterium phage BruceLethal	Pseudomonas virus LKA1
Brome mosaic virus	Propionibacterium phage PA1-14	Respiratory syncytial virus
Enterovirus J	Propionibacterium phage PAC1	Rhinovirus A
Enterovirus sp1	Propionibacterium phage PHL010M04	Rhinovirus C
Escherichia virus T4	Propionibacterium phage PHL055N00	Salicola phage CGphi29
Glossina hytrovirus	Propionibacterium phage PHL067M10	Siphoviridae Sk1virus sp1
Haloarcula virus HCIV1	Propionibacterium phage PHL150M00	Siphoviridae C2virus sp1
Halovirus HRTV-5	Propionibacterium virus ATCC29399BT	Siphoviridae Pa6virus
Hot pepper alphaendornavirus	Propionibacterium virus Attacne	Siphoviridae Sfi11virus sp1
Human coronavirus 229E	Propionibacterium virus P1.1	Siphoviridae Sfi21dt1virus sp1
Human coronavirus HKU1	Propionibacterium virus P1001	Siphoviridae sp1
Human coronavirus NL63	Propionibacterium virus P100A	Staphylococcus virus IPLAC1C
Human mastadenovirus C	Propionibacterium virus P100D	Streptococcus phage 5093
Human metapneumovirus	Propionibacterium virus P101A	Streptococcus phage EJ-1
Human orthopneumovirus	Propionibacterium virus P104A	Streptococcus phage TP-J34
Human respirovirus 3	Propionibacterium virus P105	Streptococcus virus 7201
Pandoravirus inopinatum	Propionibacterium virus PA6	Streptococcus virus 9871
Lactococcus phage BM13	Propionibacterium virus PAD20	Streptococcus virus 9872
Lactococcus phage jm2	Propionibacterium virus PHL009M11	Streptococcus virus 9874

Lactococcus phage M6165	Propionibacterium virus PHL025M00	Streptococcus virus ALQ132
Lactococcus phage phi7	Propionibacterium virus PHL060L00	Streptococcus virus Cp1
Lactococcus virus 712	Propionibacterium virus PHL071N05	Streptococcus virus DT1
Lactococcus virus ASCC191	Propionibacterium virus PHL092M00	Streptococcus virus phiAbc2
Lactococcus virus bIL67	Propionibacterium virus PHL111M01	Streptococcus virus Sfi19
Lactococcus virus c2	Propionibacterium virus PHL112N00	Streptococcus virus Sfi21
Lactococcus virus CB14	Propionibacterium virus PHL114L00	Thermus phage phiYS40
Lactococcus virus SI4	Propionibacterium virus PHL116M00	Thermus phage TMA
Myoviridae T4virus sp1	Propionibacterium virus PHL132N00	Tomato brown rugose fruit virus
Nupapillomavirus 1	Propionibacterium virus PHL141N00	Tomato mosaic virus
Orpheovirus IHUMI-LCC2	Propionibacterium virus PHL179M00	Tomato mottle mosaic virus
Orthopneumovirus sp1	Propionibacterium virus PHL301M00	Uncultured crAssphage
Pandoravirus quercus	Propionibacterium virus Procrass1	Vibrio phage VvAW1
Pepper mild mottle virus	Propionibacterium virus Solid	Zaire ebolavirus

Figure (letter only). Relationship of abundant viruses with the risk of higher severity in infants hospitalized for bronchiolitis in metatranscriptome

The violin plots show the comparison of the distribution of annotated ten most abundant viruses in the nasopharyngeal airway of infants hospitalized for bronchiolitis, according to the PPV use. Each point represents each infant. The grey bar and rectangle represent the mean and 95% confidence interval. In the violin plots, the width represents the probability that infants take on a specific relative abundance. Abbreviations: FDR, false discovery rate; PPV, positive pressure ventilation

Second, we agree with the reviewer that the strong association of *S. pneumoniae*/*S. aureus* module with severity is an important finding. Research has suggested that interactions between RSV and these species alter microbial gene expression (e.g., *ply*, *pbp1A*), thereby increasing the virulence and worsening disease severity⁶. Our finding indicates a complex interplay between respiratory viruses, these microbial species, their function, and host response in the airway and their integrated contribution to the bronchiolitis pathobiology. Our data should facilitate further investigation into the identification of causal key drivers of disease severity as well as the role of modulating microbiome (e.g., pre- and probiotics) in the treatment of severe bronchiolitis.

As suggested, we have also added these points to the Discussion section as follows: “Lastly, research has also shown that direct interactions between RSV and *S. pneumoniae* alter microbial gene expression (e.g., *ply*, *pbp1A*), thereby increasing the virulence and worsening disease severity⁶. Our inferences—in conjunction with the existent evidence—indicate a complex interplay between respiratory viruses, these microbial species, their function, and host response in the airway, and their integrated contribution to the bronchiolitis pathobiology. Our data should facilitate further investigations to disentangle the complex web and to determine the role of modulating microbiome (e.g., prebiotics and probiotics) in the treatment of severe bronchiolitis.” (page 15, para 1).

8. Given the above comment, it would have been nice to see more Discussion about how the current findings compare/contrast with prior results from this group and other cohorts. There is some discussion of this, but it feels limited. For example, in light of their earlier paper that reported different integrated endotypes of RSV infection (and risk for asthma). I acknowledge that the focus of the current manuscript is on an entirely different clinical outcome, but nonetheless it seems there may be potential overlap in results of interest for a comparative discussion. The mentioned paper is just one suggestion since RSV was the predominant virus found in the analyzed cohort here (and severe bronchiolitis in turn is associated with increased risk for asthma). But there might be relevant results from other studies that could also be considered for a more expanded Discussion. Similarly, it would be nice to see more Discussion on the microbial functional expressions identified, including mention of whether any of the functions identified could be mapped to specific bacteria. There is some such discussion (lines 148-160), however no mechanistic experiments were performed to potentially validate the computational findings. To determine, for example, if specific genes in the reported pathways were indeed up-regulated or down-regulated in specific bacteria by targeted PCR.

[Response]

We thank the reviewer for these thoughtful comments. First, our previous *Nat Commun* study³—which focused on patients with RSV infection and the asthma outcome—reported that, compared to the reference endotype (clinically resembling “classic” RSV bronchiolitis), the most-severe endotype (e.g., 19% with PPV use) also had a higher abundance of *S. pneumoniae* and unique host response profile (e.g., low type-I interferon response). While the *Streptococcus*-severity relationship is consistent, we also note that the microbial function was not examined in the previous study. Additionally, this endotype also had a non-significantly higher risk of asthma by age 5 years (OR 2.29, 95% CI 0.74-8.07). As suggested, we have added these to the Discussion section. The text now states: “Furthermore, our previous integrated omics analysis of infants with RSV bronchiolitis—which focused on the microbiome taxonomy (i.e., not function), host transcriptome, metabolome, and asthma outcome—found that the most-severe endotype

(e.g., 19% with PPV use) also had a higher abundance of *S. pneumoniae* and unique host response profile (e.g., low type-I interferon response). This endotype also had a non-significantly higher risk of asthma by age 5 years.³⁰” (page 13, para 1).

Second, the current study did not have data of targeted PCR of cultured species (without interaction with host immune response). As suggested, we have added this important point to the Limitations section as follows: “Third, the current study did not have mechanistic experiments to validate the identified microbial functions.” (page 16, para 1).

Lastly, as requested, we have expanded the discussion on the microbial function (e.g., BCAA metabolism, oxidative stress response, NADH pathways) as follows: “Second, studies have also shown the role of oxidative stress response in the virulence of microbes in the oxygen-rich environment, such as the airway⁷. For example, *S. pneumoniae* employs predominantly enzymatic mechanisms (e.g., NADH oxidase, superoxide dismutase) to eliminate the effects of oxidative stress⁷. Indeed, loss of the NADH oxidase activity encoded by *nox* results in a decrease in the virulence of *S. pneumoniae*⁸. Additionally, NADH oxidase contributes to the virulence of *S. pneumoniae* as an adhesin—an important cell-surface component in the infectious process—and elicits a protective immune response in mice⁹. Lastly, research has also shown that direct interactions between RSV and *S. pneumoniae* alter microbial gene expression (e.g., *ply*, *pbp1A*), thereby increasing the virulence and worsening disease severity⁶. page 14, para 2 to page 15, para 1).

9. There is a typo in Figure 1 related to misspelling of transcriptome.

[Response]

We thank the reviewer for the careful review. We have fixed the typo in **Figure 1**.

Figure 1. Analytic flow of integrated-omics analysis

COMMENTS FROM REVIEWER #2:

I am reviewing a manuscript by Fujiogi et al. titled “Integrated relationship of nasopharyngeal airway, host response and microbiome with bronchiolitis severity: dual-transcriptomic profiling“. I found the manuscript excellently constructed and the conclusions sound. I have a few, small recommendations and two particular criticisms. If code/data was available for review, it would greatly strengthen my confidence in the authors’ results.

[Response]

We thank the reviewer for these comments. As detailed in the response to the comment #4, we have provided access to the source data, data objects for each figure, and computational code. The source data that support the findings of this study are now available on the NIH/NIAID ImmPort through controlled access (<https://www.immport.org/shared/study/SDY1883>), according to the Data Sharing Plan approved by the NIH. We note that, in addition to the importance of data sharing, compliance with the study informed consent is required. The informed consent form of MARC study states “Your child’s samples and information will be used to study the possible genetic causes of severe bronchiolitis, recurrent wheezing, asthma and related concepts”. Thus, the datasets are available through controlled access. In addition, the computational code has been uploaded to the following GitHub website (https://github.com/HasegawaLab/sample_code_dual_transcriptome_nc_open.git).

As requested, we have revised the data availability and code availability sections as follows:

“Data availability: The data that support the findings of this study are available on the NIH/NIAID ImmPort (<https://www.immport.org/shared/study/SDY1883>) through controlled access to be compliant with the informed consent forms of the MARC-35 study and the genomic data sharing plan. Source data without participant-level data are provided with this paper.” (page 23, para 2), and **“Code availability:** Computational code from the study is available at https://github.com/HasegawaLab/sample_code_dual_transcriptome_nc_open.git.” (page 23, para 3).

1. Some of the figures need to have labels, especially the volcano plots. Some of the integrated data can be used in co-occurrence plots, I think some of the data can be plotted using cytoscape and could be a pretty good figure regarding PPV outcomes.

[Response]

We thank the reviewer for these suggestions. First, as requested, we have added the labels and changed dot colors according to the direction in **Figures 2A** and **4A** (see the following pages). Additionally, we have also created a co-occurrence plot by using the Cytoscape. This new plot (**Supplemental figure 8**; see page 22 of this letter) shows a complex web of major clinical characteristics, airway microbiome, and host immune responses in the pathobiology of infant bronchiolitis. The text now states that “A

correlation network (**Supplemental Figure 8**) suggests a complex relationship between clinical characteristics, airway microbiome, host immune responses, and severity outcome in the nasopharyngeal airway of infants with bronchiolitis.” (the Results section, page 10, para 2) and “to visualize relationship between major clinical characteristics and dual-transcriptome modules, we developed a co-occurrence plot based on the Spearman’s correlation by using Cytoscape⁷⁰.” (the Methods section, page 22 para 2 to page 23, para 1).

Figure 2A. Differential gene expression analysis of host transcriptome data with regard to the use of positive pressure ventilation in infants hospitalized for bronchiolitis

Volcano plot of differentially expressed genes (transcriptome). The threshold of log₂ fold change is |0.58| (i.e., ≥|1.5|-fold change), and that of FDR is <0.05. There were 197 differentially expressed transcripts that met these criteria.

* Up-regulated or down regulated genes are defined as log₂ fold change ≥ 1.5 and FDR < 0.05 in PPV use compared to no PPV use.

Figure 4A. Differential gene expression analysis of microbial function data with regard to the use of positive pressure ventilation in infants hospitalized for bronchiolitis

A) Volcano plot of differentially expressed microbial transcripts (metatranscriptome). The threshold of log₂ fold change is |0.58| (i.e., ≥|1.5|-fold change), and that of FDR is <0.05. There were 129 differentially expressed microbial transcripts that met these criteria.

* Up-regulated or down regulated genes are defined as log₂ fold change ≥1.5 and FDR < 0.05 in PPV use compared to no PPV use.

Supplemental Figure 8. Correlation network of major clinical variables and dual-transcriptome modules

Nodes are represented by different colors corresponding to clinical variables and module categories. Edges show correlations between two variables and/or nodes. Edges with a Pearson correlation of greater than 0.15 are shown. Positive correlations are displayed as red; negative correlations are displayed as blue. Edge thickness is proportional to the strength of the correlation.

Abbreviations: ATP, adenosine triphosphate; BCAA, branched-chain amino acid; FDR, false discovery rate; GPCR, G-protein-coupled receptor; HR, host response; IFN, interferon; IL, interleukin; NADH, nicotinamide adenine dinucleotide hydrogen; MC, microbial composition; MF, microbial

function

2. I would say that nasopharyngeal microbiome is not a lower airway microbiome surrogate, there are some publications that make reference that upper airway microbiome is equivalent to the lower airway microbiome, but generally this is not a widely accepted view and still controversial. I would suggest that the authors change the sentence in strengths/weakness section.

[Response]

We thank the reviewer for the comment. As suggested, we have highlighted this point in the Limitations section as follows: “Our study is based on nasopharyngeal specimens, because the use of upper airway specimens is preferable as lower airway sampling (e.g., bronchoscopy) would be invasive in these young infants... Studies in adults have reported similar but distinct microbial communities between concurrently sampled upper and lower airway specimens^{51–53}.” (page 15, para 2 to page 16, para 1).

3. How did the authors take into account time to PPV use? This was probably the most clinically oriented question I had that did not get addressed in the data or results. Was there any lead time or length time bias involved with how / when the subjects required PPV? I wonder if tit would strengthen their observations or transcriptomic signatures by taking into account time to PPV?

[Response]

We note that we do not have the exact information on the exact time interval from specimen collection (within 24 hours of hospitalization) to PPV use. As suggested, we have acknowledged this point in the Limitations section. The text now states: “Fourth, our inferences may be biased due to the relationship between the timing of treatments, specimen collections, and PPV use despite that the specimens were collected within a short time period.” (page 16, para 1).

4. Lack of code/data for review is the manuscripts greatest weakness. For a largely bioinformatic manuscript, the code should be available for review. While I can understand that raw fastq data is not available, efforts should be made for the microbiota data to be available on NIH SRA. I would recommend that data objects be generated and uploaded to a repository such as github if raw data will be going to dGAP. The data objects can contain the annotated gene/transcriptome/metatranscriptome without the raw data. Code should contain enough information to be able to generate the figure from the data objects.

<https://www.springernature.com/gp/authors/research-data-policy/data-availability->

statements/12330880) (Mirzayi et al. Nature Medicine 2021) (Langille et al. Microbiome 2018).

[Response]

We thank the reviewer for these comments. As briefly mentioned above, as requested, we have provided access to the source data, data objects for each figure, and computational code. The source data that support the findings of this study are now available on the NIH/NIAID ImmPort through controlled access (<https://www.immport.org/shared/study/SDY1883>), according to the Data Sharing Plan approved by the NIH/NIAID. We note that, in addition to the importance of data sharing, compliance with the study informed consent is required. The informed consent form of the MARC study states “Your child’s samples and information will be used to study the possible genetic causes of severe bronchiolitis, recurrent wheezing, asthma and related concepts”. Thus, the datasets are available through controlled access. Second, to be compliant with the informed consent forms of the MARC-35 study, we are precluded from publicly sharing the participant-level data, such as age, sex, and outcomes. Therefore, we have removed the participant’s level data from the data objects. Therefore, we have included only the *aggregate* data object for the figures. Third, the computational code has been uploaded to the following GitHub website (https://github.com/HasegawaLab/sample_code_dual_transcriptome_nc_open). As requested, we have revised the data availability and code availability sections as follows: “**Data availability:** The data that support the findings of this study are available on the NIH/NIAID ImmPort (<https://www.immport.org/shared/study/SDY1883>) through controlled access to be compliant with the informed consent forms of MARC-35 study and the genomic data sharing plan. Source data without participant-level data are provided with this paper.” (page 23, para 2), and “**Code availability:** Computational code from the study is available at https://github.com/HasegawaLab/sample_code_dual_transcriptome_nc_open.” (page 23, para 3).

I will be using the page PDF page numbers and the leftmost numbers

Abstract

No recommendations

Introduction

Page 5, Line 22

Figure 1 is an excellent figure and certainly signposts the goals of the manuscript. I wanted to thank the authors for including a figure to describe their analysis and conclusions.

[Response]

We thank the reviewer for the positive comment.

Results

Page 7, Line 29

Figure 2A please apply the labels to the R generated figure – I believe it should can be layered in. Also, I would suggest to label the directions (e.g., upregulated in PPV use vs. upregulated in subjects w/o PPV use).

[Response]

We thank the reviewer for these helpful comments. As we have responded to comment #1, we have revised the figure accordingly.

Page 8, Line 42

Figure 3A This is good figure, but the phylum legend has mismatched colors, it's a bit distracting.

[Response]

We have used the same colors both in the legend and the plot. The reason why they appear different is the transparency factor. Regardless, as requested, we have changed the color on the legend of **Figure 3A** (see below).

Figure 3A. Relationship of abundant microbial species with the risk of higher severity in infants hospitalized for bronchiolitis

Page 8, Line 39

I also found it interesting that the presence of one bacterial species would have divergent results, such as having Haemophilis as an increase of LogFold change as a predictor for PPV use, and lower Logfold change associated with intensive care use? Were the criteria different? Another taxa is Staph epi and pneumococcus which also shows differences between LogFold change in PPV use and intensive care use. I would have expected that increase in LogFold Change in these taxa predicted negative results.

[Response]

We appreciate the opportunity to clarify this point. We agree with the reviewer that these are intriguing. We note that these two severity outcomes are related but not necessarily identical. In general, the indication for PPV is more specific and less variable among hospitals. In contrast, intensive care use (e.g., ICU admission) is known to be *widely* variable across hospitals in the U.S. due to the difference in resource and admission criteria⁵. Therefore, we have used PPV use as the primary outcome. As suggested, we have clarified this point in the Methods section as follows: “We used PPV use as the primary outcome as it is considered more specific than intensive care use⁶⁸.” (page 21, para 1).

Page 8, Line 52

Figure 4A would benefit from the same recommendations re: Figure 2A.

[Response]

Please see our response to comment #1.

Page 9, Line 60-67

I am wondering if the investigators could demonstrate the correlations with something like cytoscape to show correlations between microbes, microbial functions, pathways, clinical factors, and outcomes. It would be a nice way to summarize otherwise busy tables. The information is present, but

[Response]

Please see our response to comment #1 and **Supplemental figure 8.**

Discussion:

Page 11, Line 105

“To the best of our knowledge, ...” I think this is a particular strength of the manuscript

with this excellent multi-omic evaluation of a sample cohort of pediatric subjects with and without PPV ventilation. One of my particular interests or next steps that the authors could do is to use the observations in a validation cohort? I wonder if this would make the manuscript stronger if there was another cohort which the authors could test their observations and predict those subjects who may go onto require PPV.

[Response]

We thank the reviewer for the helpful comment. To date, the MARC-35 study is the only large-scale cohort that has comprehensive airway data (e.g., dual-transcriptome) in infants with severe bronchiolitis. Enrolling a large number of these infants is a major challenge. These make external validation difficult. Regardless, we completely agree with the reviewer on the importance of validating our inference. As suggested, we have highlighted this point in the Discussion section. The text now states: “Fifth, while this study derives novel and well-calibrated hypotheses that facilitate future experiments, our findings warrant further external validation.” (page 16, para 2).

Page 14, Line 163

The strengths and weaknesses of the manuscript are well thought out. I am particularly impressed with the argument for nasopharyngeal testing. I would however cautiously agree with nasopharyngeal testing is equivalent to lower airway testing “...upper airway sampling provides a reliable representation ...” (line 168) I think simply stating that NP testing can predict some understanding of the lung microbiome/transcriptome without drawing equivalence would probably be a better way of stating why we should consider NP testing vs. lung testing.

[Response]

Please see our response to comment #2. As suggested, we have acknowledged this point in the Limitations sections as follows: “bronchiolitis involves inflammation of both upper and lower airways, while our study is based on nasopharyngeal specimens... studies in adults have reported similar but distinct microbial communities between concurrently sampled upper and lower airway specimens⁵¹⁻⁵³.” (page 15, para 2 to page 16, para 1).

Methods:

Page 16, Line 207

What mechanism of random selection was used to choose the infants for -omic profiling?

[Response]

We appreciate the opportunity to clarify this point. The current study used data from 244 infants who were randomly selected from the MARC-35 overall cohort (n=1016) (as shown below) by

generating random numbers using R (R Foundation, Vienna, Austria). To clarify this point, we have shown the study flow diagram in **Supplemental Figure 1** and cited it in the Results (page 7, para 1) and Methods (page 18, para 1) sections.

Supplemental figure 1. Study flow diagram

The differences in the analytic and non-analytic cohorts are summarized in **Table E1**.

* The transcriptome and metatranscriptome data are obtained in 244 infants who were *randomly* selected from the overall cohort

Page 17, Line 22

Timing of respiratory failure? I am not sure if this was clearly described, but when did the subjects require PPV ? Did timing of respiratory failure (requiring PPV) impact the multi-omic results? How was time to respiratory failure accounted for in the data/results?

[Response]

Please see our response to comment #3.

Page 21, Line 317 and 320

Data that does not include host (human data) such as the microbiome/metatranscriptome

should be uploaded to the NIH SRA. If there is concern, the authors could filter any possible human contaminant DNA/RNA and upload that data to the NIH SRA. If the reviewers are concerned re: privacy the SRA can generate a reviewer link for review (thus private). This has not been an issue with other publications. If possible, could the authors provide more information regarding the controlled access? (Will they be uploading the data through dGAP?)

[Response]

Please see our response to comment #4. The source data that support the findings of this study are now available on the NIAID ImmPort (<https://www.immport.org/shared/study/SDY1883>), according to the Data Sharing Plan approved by the NIH/NIAID.

Code availability, it is unacceptable that the code is unavailable for review. While understandable, there is no suggestion that any of the code is proprietary, thus it should be shared for review. Given that a significant amount of their work is bioinformatics/code this should be uploaded to a code repository. The data objects generated in R could be uploaded into github (this could avoid issues with raw fastq data) and code to generate the figures.

[Response]

Please see our response to comment #4. The computational code from the study is available at https://github.com/HasegawaLab/sample_code_dual_transcriptome_nc_open.git.

References

1. Tsao, C. W. & Vasan, R. S. Cohort Profile: The Framingham Heart Study (FHS): Overview of milestones in cardiovascular epidemiology. *Int. J. Epidemiol.* **44**, 1800–1813 (2015).
2. Sudlow, C. *et al.* UK Biobank: An open access resource for identifying the causes of a wide range of complex diseases of middle and old age. *PLoS Med.* **12**, 1–10 (2015).
3. Raita, Y. *et al.* Integrated omics endotyping of infants with respiratory syncytial virus bronchiolitis and risk of childhood asthma. *Nat. Commun.* **12**, 3601 (2021).
4. Fujiogi, M. *et al.* Trends in bronchiolitis hospitalizations in the United States: 2000-2016. *Pediatrics* **144**, e20192614 (2019).
5. Mansbach, J. M. *et al.* Prospective multicenter study of children with bronchiolitis requiring mechanical ventilation. *Pediatrics* **130**, e492-e500 (2012).
6. Smith, C. M. *et al.* Respiratory syncytial virus increases the virulence of streptococcus pneumoniae by binding to penicillin binding protein 1a a new paradigm in respiratory infection. *Am. J. Respir. Crit. Care Med.* **190**, 196–207 (2014).
7. Bortoni, M. E., Terra, V. S., Hinds, J., Andrew, P. W. & Yesilkaya, H. The pneumococcal response to oxidative stress includes a role for Rgg. *Microbiology* **155**, 4123–4134 (2009).
8. Schurig-Briccio, L. A. *et al.* Role of respiratory NADH oxidation in the regulation of Staphylococcus aureus virulence. *EMBO Rep.* **21**, 1–15 (2020).
9. Muchnik, L. *et al.* NADH Oxidase Functions as an Adhesin in Streptococcus pneumoniae and Elicits a Protective Immune Response in Mice. *PLoS One* **8**, (2013).

REVIEWERS' COMMENTS

Reviewer #1 (Remarks to the Author):

The comments/queries for the initial review have been well addressed by the authors. The manuscript is much improved in transparency and clarifications, and the additional analyses undertaken provide novel data and insights for the field.

Reviewer #2 (Remarks to the Author):

The authors have contributed significant work and their manuscript is stronger. I have no further concerns regarding their work.